# FEATHER: Lifelong Test-Time Adaptation with Lightweight Adapters

## Abstract

Lifelong/continual test-time adaptation (TTA) refers to the problem where a pre-trained source domain model needs to be continually adapted at inference time to handle non-stationary test distributions. Continuously updating the source model over long horizons can result in significant drift in the source model, forgetting the source domain knowledge. Moreover, most of the existing approaches for lifelong TTA require adapting all the parameters, which can incur significant computational cost and memory consumption, limiting their applicability on edge devices for faster inference. We present FEATHER (liFelong tEst-time Adaptation wiTH lightwEight adapteRs), a novel lightweight approach that introduces only a small number of additional parameters to a pre-trained source model which can be unsupervisedly and efficiently adapted during test-time for the new test distribution(s), keeping the rest of the source model frozen. FEATHER disentangles the source domain knowledge from the target domain knowledge, making it robust against error accumulation over time. Another distinguishing aspect of FEATHER is that, unlike some recent approaches for lifelong TTA that require access to the source data for warm-starting the adaptation at test time, FEATHER does not have such a requirement. FEATHER is also orthogonal to the existing lifelong TTA approaches and can be augmented with these approaches, resulting in a significant reduction in the number of additional parameters needed to handle the lifelong TTA setting. Through extensive experiments on CIFAR-10C, CIFAR-100C, ImageNetC, and ImageNet3DCC Robustbench benchmark datasets, we demonstrate that, with substantially (**85% to 94%**) fewer trainable parameters, FEATHER achieves better/similar performance compared to existing SOTA lifelong TTA methods, resulting in faster adaptation and inference at test-time. The source code for FEATHER will be released upon publication.

## 1 Introduction

Real-world applications of deep learning models routinely encounter test data that may come from a non-stationary distribution, that is different from the source training data distribution. For example, when deployed in the wild, a model trained on clean images may observe various kinds of domain shifts, such as low-light situations, camera flares, etc., at test time. In such settings, the source pre-trained model is required to adapt at test (inference) time without any access to any labeled data from the test domain. This problem setting is known as *test-time adaptation* (TTA) (Liang et al., 2023; Sun et al., 2020; Liang et al., 2020; Liu et al., 2021; Wang et al., 2021; Zhou & Levine, 2021; S & Fleuret, 2021). Moreover, doing so in a setting when the test domain itself may continuously undergo a shift over time is even more challenging; in this setting, we need to ensure that the model performs well on the new domain(s) while also not suffering from forgetting on the previously seen domains in order to maintain its predictive accuracy on test inputs from previous domains. This problem setting is referred to as *lifelong/continual* TTA and has received significant recent interest (Wang et al., 2021; Niu et al., 2022; Hong et al., 2023; Song et al., 2023).

Although recent lifelong TTA methods have shown strong performance, they incur significant overheads at adaptation/inference time (e.g., the adaptation typically requires updates to all the parameters) and memory consumption. They usually also require access to the source domain data (not practical in many settings) for warm-starting the adaptation for the test domains. Moreover, despite using mechanisms to control the forgetting of the knowledge of the source domain, when faced with long horizons of test distribution shifts, these methods may still suffer from significant forgetting.

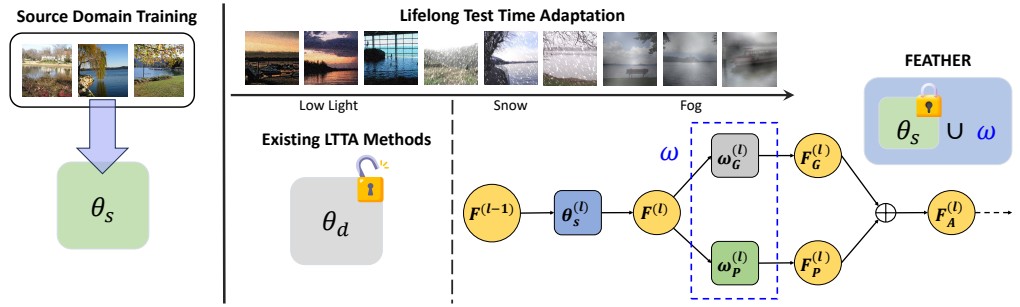

Figure 1: An overview of the lifelong/continual TTA setting, where a model trained on the source domain (represented as clean images) is adapted to different domains occurring during test time (Low Light, Snow, Fog, etc.) sequentially without any labels. Existing approaches adapt the same model to the target domain, losing the source knowledge ($\theta_s$). In contrast, our proposed method FEATHER (shown on the right) adapts the newly added parameters ($\omega$), keeping the source knowledge intact. Refer to Section 3.1 for more details.

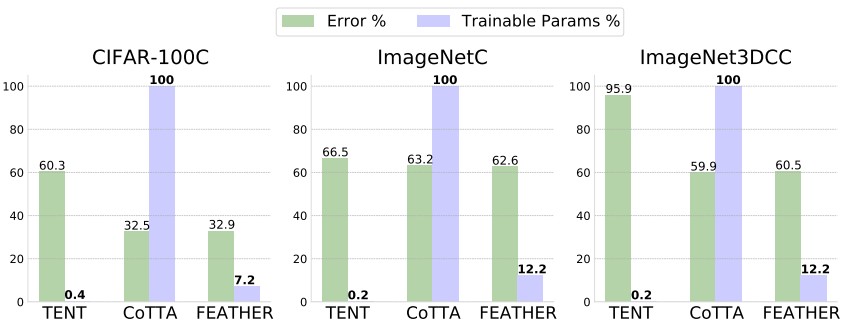

Figure 2: Error and % of trainable parameters for CIFAR100C, ImageNetC, and ImageNet3DCC. Lower is better for both error and trainable parameter percentage. TENT, adapting only BN params, leads to the lowest number of parameter updates during test time; however, the error accumulation in TENT results in poor performance (high error rate). CoTTA adapting the entire model during test time (100% trainable parameters) significantly improves the error rate. In contrast, FEATHER (adapting only newly added adapter parameters) performance shows that a similar error rate can be achieved with a drastic reduction in trainable parameters (**only 7.2% and 12.2% trainable parameters**), making the TTA methods parameter efficient.

In this work, we propose FEATHER (li**F**elong t**E**st-time **A**daptation wi**TH** lightw**E**ight adapte**R**s), a parameter-efficient approach, to address both these issues in a principled way. In particular, the design of FEATHER (Fig. 1) ensures that (1) it does not require adapting all the model parameters at test time while still resulting in performance that is better/comparable to approaches that require updating all the parameters, and (2) unlike existing continual TTA approaches based on pseudo-labeling and entropy-minimization (Wang et al., 2021), which are prone to error accumulation over time, FEATHER disentangles the source (training) and target (test) domain knowledge so that even with continuous shifts in the test domain, the knowledge of the source domain remains preserved, enabling domain-specific TTA for each future domain. FEATHER achieves this by introducing a set of lightweight adapters to a base architecture. These adapters can be efficiently updated at inference time (with the base architecture's weights remaining frozen) using only the unlabeled test data. Another distinguishing aspect of FEATHER is that the updates to the adapter parameters does not require access to the source domain data (which some recent works require in order to do a warm-start (Song et al., 2023)). We evaluate FEATHER on four widely used benchmarks in lifelong TTA (CIFAR10C, CIFAR100C, ImageNetC, and ImageNet3DCC) and show that the existing methods can be made parameter efficient by a massive margin (**85%-94% fewer trainable parameters**; also see Fig. 2) with comparable or better performance on the task.

## 2 PROBLEM SETUP AND FORMULATION

Let $\mathcal{D}_\mathcal{S} = \left\{ \mathbf{x}_s^{(m)}, y_s^{(m)} \right\}_{m=1}^M$ denote the source domain ($\mathcal{S}$) data used to train a model $f_\theta$ where $\theta$ denotes the model parameters. The model $f_\theta$ takes an input $\mathbf{x}$ and makes a prediction $\hat{y}$ (the

predicted label corresponds to the index of the maximum value in the output vector). We denote the source domain pre-trained parameters as $\theta_s$. During the training phase, the model $f_\theta$ is pre-trained using source domain data as $\theta_s = \min_\theta \frac{1}{M} \sum_{m=1}^M \mathcal{L}_\mathcal{S} \left( f_\theta \left( \mathbf{x}_s^{(m)} \right), y_s^{(m)} \right)$, where $\mathcal{L}_\mathcal{S}$ is the supervised loss function for the source domain $\mathcal{S}$ data, and $y_s^{(m)}$ is the ground truth label for the input $\mathbf{x}_s^{(m)}$. An example of the supervised loss function for classification is the cross-entropy loss: $\mathcal{L}_\mathcal{S}(f_\theta(\mathbf{x}_s), y_s) = - \sum_k y_s^{(k)} \log f_\theta(\mathbf{x}_s)^{(k)}$, where the index $k$ denotes the $k^{th}$ class.

At test-time, we assume access to only the source domain pre-trained model $\theta_s$ and do not assume access to source domain data. In test-time adaptation, the goal is to adapt the source model to predict the labels of unlabeled test inputs from a different distribution. Formally, let $\mathcal{D}_{\mathcal{T}_d} = \left\{ \mathbf{x}_d^{(n)} \right\}_{n=1}^{N_d}$ denote the test data from the target domain $\mathcal{T}_d$. We assume no access to $\mathcal{D}_\mathcal{S}$ during test-time and are only provided with $\theta_s$. At test time, for a domain $\mathcal{T}_d$, we perform an adaptation $\theta_s \to \theta_d$ using the test inputs $\mathcal{T}_d$ to get adapted parameters $\theta_d$ as $\theta_d = \min_\theta \frac{1}{N_d} \sum_{n=1}^{N_d} \mathcal{L}_{\mathcal{T}_d} \left( f_\theta \left( \mathbf{x}_d^{(n)} \right) \right)$, where $\mathcal{L}_{\mathcal{T}_d}$ is the test-time adaptation loss for test inputs, that can be realized using an unsupervised loss. For example, one such unsupervised TTA loss for classification used by TENT (Wang et al., 2021) is the entropy loss $\mathcal{L}_{\mathcal{T}_d}(f_\theta(\mathbf{x}_d)) = - \sum_k f_\theta(\mathbf{x}_d)^{(k)} \log f_\theta(\mathbf{x}_d)^{(k)}$, where the $k$ denotes the class index. Other losses, such as cross-entropy between a student and teacher model, can also be employed (Wang et al., 2022). Having obtained $\theta_d$, we make predictions for a test input $\mathbf{x}_d$ from the new domain as $\hat{y} = f_{\theta_d}(\mathbf{x}_d)$.

In the continual/lifelong TTA setting (Wang et al., 2022; Song et al., 2023), the test examples can arrive sequentially from different target domains. Thus, in lifelong TTA, there can be $D$ target domains $\{\mathcal{T}_d\}_{d=1}^D$, making the distribution of test data non-stationary. Moreover, we assume that the learner gets no information about the switch in the domain. These aspects make lifelong TTA considerably more challenging than standard TTA. Further, in *online* lifelong TTA, the learner gets to see the test input only once, and multiple passes are not allowed.

To handle the above issues and the error accumulation and catastrophic forgetting of the earlier domains caused by the continuously drifting test domain distributions, we present a framework based on augmenting a base network with a small number of trainable adapter parameters (Houlsby et al., 2019; Hu et al., 2021; Varshney et al., 2021). The base network is kept frozen at test time, and only the adapter parameters are updated using an unsupervised loss. Updating only the adapters significantly improves the parameter efficiency without compromising the performance. We would like to note here that, while the idea of adapters has been used for efficient supervised finetuning and continual learning, we leverage adapters for the lifelong TTA setting where we are required to perform *unsupervised* finetuning of the model at test time.

## 3 LIFELONG TEST-TIME ADAPTATION WITH ADAPTERS

In this section, we provide a detailed description of our framework. While our framework is general and can be applied to a variety of base architectures and adapters for vision as well as NLP tasks, here we describe it assuming a feed-forward base architecture in which we employ group-wise and point-wise convolutional filters between layers (as shown in Fig. 1). These additional filters, which consist of only a small number of additional parameters, act as adapters, and can be efficiently updated at test time given unlabeled input(s) from a new distribution.

### 3.1 **FEATHER**: LIFELONG TEST-TIME ADAPTATION WITH LIGHTWEIGHT ADAPTERS

One approach to solving TTA is to update the entire network parameters at test time. However, this can lead to forgetting of the knowledge of the source domain as well as (in the continual TTA setting) the knowledge of other previously encountered domains. To address this issue, recent work has considered keeping the source model frozen except for the batch normalization parameters (Wang et al., 2021), or simply updating the batch normalization statistics using the test data (Schneider et al., 2020). A drawback of this approach is its reduced flexibility/capacity in handling test distributions that might be significantly different from the source domain distribution.

Another line of work adapts the entire network, but to mitigate forgetting, they introduce a parameter restoring mechanism that resets some of the parameters back to the source domain pre-trained

model (Wang et al., 2022; Brahma & Rai, 2023). Even though the resetting mechanism is designed to handle error accumulation in the long run, updating the entire network can still result in error accumulation over time. Moreover, updating the entire network and restoring back by a small percentage at every update iteration makes the adaptation and inference computationally heavy since the gradients need to be computed for the entire model parameters with respect to the minibatch.

In contrast to these approaches, our approach FEATHER introduces lightweight adapters to the base network and, at test time, only updates the adapter parameters using an unsupervised loss. Adapting additional weights not only helps reduce the computational overhead drastically but also provides control over the error accumulation due to a small number of adapted parameters.

When adapting the optimal source domain parameters $\theta_s$ to the optimal target domain parameters $\theta_d$, we also wish to preserve the source domain knowledge (represented by $\theta_s$). To achieve this, we rely on updating only the adapter parameters, which we denote as $\omega$, while keeping $\theta_s$ frozen at test time. Therefore, the parameters $\theta_d$ of a test domain $d$ would be $\theta_s \cup \omega$.

For the choice of a lightweight adapter, the primary objective is parameter efficiency, with comparable or better predictive accuracy. Previously, a wide range of design choices have been proposed to make the adapters parameter efficient (Houlsby et al., 2019; Hu et al., 2021; Varshney et al., 2021). In our work, we specifically design adapters considering the requirement for making them compatible with identity transformation to remove the dependency on the availability of the source training dataset for initial warmups (see section 3.2 for more details). Primarily, given a pre-trained model (also referred to as *base model* in TTA setup) with parameters $\theta_s$, we insert new adapter parameters ($\omega$) in between layers. Considering every layer in the base model ($\theta_s^{(l)}$) acting as a sequence of feature transformations over the input, we insert adapters ($\omega_s^{(l)}$) after the feature transformations. For instance, consider a sequence of transformations present in the base model

$$\mathbf{F}^{(l-1)} \to \theta_s^{(l)} \to \mathbf{F}^{(l)} \to \ldots \mathbf{F}^{(l+n-1)} \to \theta_s^{(l+n)} \to \mathbf{F}^{(l+n)}$$

where $\mathbf{F}^{(l)}$ represents the transformed features after the $l^{th}$ layer of the base model ($\theta_s^{(l)}$). Note $\mathbf{F}^{(l)} \in \mathbb{R}^{h \times w \times c}$ here denotes the feature map with $h$, $w$ and $c$ as its the height, width, and number of channels, respectively, where the parameters $\theta_s^{(l)}$ define a convolution operation $g_{\theta_s^{(l)}}(\mathbf{F}^{(l-1)})$. After inserting adapters in between, we obtain the sequence

$$\mathbf{F}^{(l-1)} \to \theta_s^{(l)} \to \mathbf{F}^{(l)} \to \omega^{(l)} \to \mathbf{F}_\omega^{(l)} \to \ldots \mathbf{F}^{(l+n-1)} \to \theta_s^{(l+n)} \to \mathbf{F}^{(l+n)}$$

where $\mathbf{F}_\omega^{(l)}$ depicts the transformation made by the newly added adapters. Fig.1 (lower right) provides a detailed representation of such a sequence. In practice, we only insert the adapters in a few locations, depending on the computational and memory budget. For test-time adaptation, we propose to adapt only the newly added adapter parameters, keeping the rest of the network frozen. For the $l^{th}$ layer, we denote the frozen parameters and adapter parameters using $\theta_s^{(l)}$ and $\omega^{(l)}$, respectively. For brevity of notation, we omit $l$, and use $\theta_s$ and $\omega$ to denote $\theta_s^{(l)}$ and $\omega^{(l)}$, respectively.

The choice of adapters is architecture-specific, with their parameter-efficiency and ease of parameter update being the key consideration. For example, for language tasks, one choice could be low-rank adapters, such as LoRA (Hu et al., 2021). In this work, we focus on vision tasks with convolution-based architectures. For this setting, we propose using a combination of pointwise and groupwise convolution for adapter modules, which can be used as lightweight, parameter-efficient adapters. Groupwise COnvolution (GCO) using $r$ number of groups, where $r \ll c$ reduces the number of parameters by a considerable margin, requiring only $\frac{c}{r}$ times fewer parameters than the standard convolution filter. In contrast, PointWise Convolution (PWC) helps handle the drawback of GCO capturing fewer feature maps. For PWC, we use convolution filters of size $1 \times 1 \times c$, which are $9\times$ more parameter efficient than standard $3 \times 3$ convolution operation. Combining both operations (GWC and PWC) makes the transformation parameter efficient by a significant margin.

We use $g_{\omega_G}$ to denote $3 \times 3$ GCO operation of group size $r$ having adapter parameters $\omega_G$ and $g_{\omega_P}$ to denote the PWC operation having adapter parameters $\omega_P$. GCO and PWC modify the feature map as $\mathbf{F}_G^{(l)} = g_{\omega_G}(\mathbf{F}^{(l)})$ and $\mathbf{F}_P^{(l)} = g_{\omega_P}(\mathbf{F}^{(l)})$, respectively. Further, we use $\omega$ to collectively denote $\omega_G$ and $\omega_P$. A noteworthy point about the proposed mechanism is that the base model architecture needs no modifications for insertion of these adapters since the dimensions of $\mathbf{F}_G^{(l)}$ and $\mathbf{F}_P^{(l)}$ are the

same as the incoming feature map $\mathbf{F}^{(l)}$. Combining $\mathbf{F}_{\mathrm{G}}^{(l)}$ and $\mathbf{F}_{\mathrm{P}}^{(l)}$, we obtain the final transformed feature map, $\mathbf{F}_{\mathrm{A}}^{(l)}$, using adapters as $\mathbf{F}_{\mathrm{A}}^{(l)} = \mathbf{F}_{\mathrm{G}}^{(l)} \oplus \mathbf{F}_{\mathrm{P}}^{(l)}$, where $\oplus$ denotes element-wise addition. (also see Fig. 1, lower right, for a visual representation of the proposed operation)

## 3.2 PRESERVING SOURCE KNOWLEDGE WITH ZERO AND IDENTITY INITIALIZATION

Introducing adapters after any layer of the pre-trained source model can affect the feature representations of source data. Therefore, an initialization scheme is required for the adapter parameters to ensure that the source feature representations are not affected. One way to address this issue is to initialize the adapter parameters using a warmup training done on the source dataset (Song et al., 2023). However, this requires access to the source dataset. In FEATHER, we address this issue by introducing a new initialization strategy which avoids the need of access to the source data.

Specifically, we make the initial configuration of the adapter parameters equivalent to an identity function. The parameters of GWC ($\omega_{\mathrm{G}}$) are initialized with zeros. For PWC parameters ($\omega_{\mathrm{P}}$), we initialize it with a four-dimensional tensor $\mathcal{Q}$ (input channel $\times$ output channel $\times$ kernel size $\times$ kernel size) where kernel size is 1, and the first two dimensions reflect an identity matrix for no interaction between the channels. With this initialization, for the initial configuration, we have $\mathbf{F}_{\mathrm{G}}^{(l)} = g_{\omega_{\mathrm{G}} \leftarrow \mathbf{0}}\left(\mathbf{F}^{(l)}\right) = \mathbf{0}$, and $\mathbf{F}_{\mathrm{P}}^{(l)} = g_{\omega_{\mathrm{P}} \leftarrow \mathcal{Q}}\left(\mathbf{F}^{(l)}\right) = \mathbf{F}^{(l)}$, and the overall transformation due to the adapter becomes $\mathbf{F}_{\mathrm{A}}^{(l)} = \mathbf{F}_{\mathrm{G}}^{(l)} \oplus \mathbf{F}_{\mathrm{P}}^{(l)} = \mathbf{0} \oplus \mathbf{F}^{(l)} = \mathbf{F}^{(l)}$, which shows that our proposed initialization of adapters makes the initial adapter parameter configuration equivalent to an identity function. Appendix D Fig. 3 elaborates on the activation space of the PWC kernel applied on the input feature map, leading to no cross-channel interactions and the same output feature map as input.

## 3.3 PARAMETER UPDATES FOR THE ADAPTERS

Note that, for FEATHER, only the adapter parameters ($\omega$) are trainable and the rest of the network parameters ($\theta_s$) remain frozen as the source domain pre-trained weights. This ensures disentanglement of the source domain knowledge and the target domain knowledge, prevents forgetting the source domain knowledge, and the trainable $\omega$ parameters can continually acquire knowledge from the dynamically changing target domains. Since the learner is agnostic to the change in the domain, we adapt the adapter parameters using examples from a test input batch $\mathbf{x}_b$ from time step $t$ to $t+1$ as $\omega_t \rightarrow \omega_{t+1}$, which is done as $\omega_{t+1} = \omega_t - \eta \nabla_\omega \mathcal{L}_{\mathcal{U}}(f_\omega(\mathbf{x}_b))$, where $\eta$ is the learning rate, $\mathbf{x}_b$ is a test input batch from domain $d$, $f_\omega$ is the model where $\theta_s$ parameters are frozen and only the adapter parameters $\omega$ are learnable, and $\mathcal{L}_{\mathcal{U}}(f_\omega(\mathbf{x}_b))$ is the learning objective defined with respect to adapter parameters ($\omega$), which can be any unsupervised test-time adaptation loss.

## 4 RELATED WORK

**Test-Time Adaptation (TTA):** There has been significant recent progress on the problem of test-time adaptation Liang et al. (2023). Test entropy minimization (TENT) (Wang et al., 2021) adapts the batch-normalization (BN) parameters utilizing entropy minimization for test data predictions. (Schneider et al., 2020) proposes a method to perform test-time adaptation by altering the source domain's batch normalization (BN) statistics using the statistics obtained from the test inputs. EATA (Niu et al., 2022) addresses TTA by employing a weight regularizer; however, it primarily emphasizes on preventing model forgetting of the source knowledge in TTA and does not specifically cater to the challenges associated with forgetting in *lifelong* TTA. Niu et al. (2023) propose sharpness-aware entropy minimization and batch-agnostic (group or layer) norm for TTA under wild test settings. Chen et al. (2023) utilizes a learnable consistency loss, introducing adaptive parameters after each block, and only updates them during test-time. However, the effectiveness of their proposed adaptive parameters is limited to addressing multi-source and single-source domain generalization tasks for a non-continual setting, and their focus is not on parameter efficiency.

**Lifelong/Continual Test-time Adaptation:** CoTTA (Wang et al., 2022) addresses the challenge of online lifelong Test-Time Adaptation (TTA) by utilizing weight averaging and augmentation averaging techniques, as well as randomly restoring parameter values to the source domain model parameters. NOTE (Gong et al., 2022) tackles the challenge of adapting to dynamic target domains by including a normalization layer to handle instances that fall out of distribution and store the simulated i.i.d. data in memory obtained using balanced reservoir sampling. Gan et al. (2023) utilizes

image-level visual prompts for adapting to target domains, keeping the source model parameters intact. MECTA (Hong et al., 2023) performs pruning on cache data for back-propagation leading to a reduction in memory requirement. Thus, MECTA is orthogonal to the parameter-efficient approach of making lifelong TTA efficient with respect to the number of trainable parameters. EcoTTA (Song et al., 2023) utilizes meta networks to adapt the frozen original network to the target domain and a self-distilled regularization to handle catastrophic forgetting and error accumulation. However, the main drawback of EcoTTA is the requirement of source domain training data that is needed in the warm-up process of the meta-networks.

## 5 EXPERIMENTS

We evaluate FEATHER on several benchmark datasets which include CIFAR10C, CIFAR100C, ImageNetC, and ImageNet3DCC, and compare it with relevant baselines. For fairness of comparison, our baselines consist of methods that use the same training objective/mechanism and do not assume access to the source domain training data at test time.

There are multiple corruptions in a benchmark dataset and the learner comes across a test input batch remaining agnostic to the information about which domain this batch has come from. For instance, CIFAR10C and CIFAR100C consist of images from 15 different types of image corruptions that can occur due to reasons such as adverse weather conditions, low light, camera aberration, etc. More details of the benchmark datasets are provided in Appendix B.

**Evaluation Metrics:** For evaluation metrics, we follow existing approaches and report the error rate. We also compute negative log-likelihood (NLL) and Brier score to compare the uncertainty estimates of the approaches. Details of all the evaluation metrics are present in Appendix C. For computational complexity and parameter efficiency measures, we use the number of trainable/adaptable parameters along with GPU memory budget and wall-clock time.

### 5.1 COMPARED APPROACHES

In order to evaluate the efficacy of FEATHER, we conduct a comparative analysis of its performance against several (lifelong) test-time adaptation approaches. *Source* indicates the source domain pre-trained model without any adaptation. *Pseudo-label* (Lee et al., 2013) utilizes hard pseudo-labels and updates the batch normalization parameters using backpropagation. *BN Adapt* (Li et al., 2017; Schneider et al., 2020) only computes the batch normalization statistics while keeping all the network parameters frozen, including the Batch Norm parameters. *TENT-online* (Wang et al., 2021) denotes the performance of TENT in the setting when the test data arrives continually, but the information about domain change is accessible. This knowledge about change in the domain makes the learning problem much simpler. Nonetheless, such information regarding the change in the domain may not be readily available in practical situations. *TENT-lifelong* indicates the performance of TENT in the lifelong TTA setting, where the domain change information is unavailable. *CoTTA* (Wang et al., 2022) utilizes weight-averaged, augmentation averaged pseudo labels and random restoration of a small part of parameters to the source pre-trained parameters. Apart from these baselines, in Table 4, we also report some additional comparisons of FEATHER with other recent SOTA methods, such as NOTE (Gong et al., 2022), EATA (Niu et al., 2022) and EcoTTA (Song et al., 2023).

### 5.2 RESULTS

For lifelong/continual test-time adaptation, Table 1-3 summarizes our results on the 4 benchmark datasets where we compare FEATHER with other methods. For all the experiments with FEATHER, we use the learning objective and TTA scheme proposed by CoTTA Wang et al. (2022). In every TTA setting, the model pre-trained on the source dataset is termed the base model ($\theta_s$). CoTTA unfreezes all the model parameters and adapts these parameters during test time. In contrast, FEATHER adds the proposed lightweight adapters to the pre-trained base model and only adapts the newly added adapter parameters ($\omega$) along with the BN parameters of the base model. Note that the primary objective of FEATHER is to reduce the parameter update cost while maintaining the adaptation performance. Since the newly added parameters are inserted in between layers, the added adapter modules ensure equal input and output dimensions at the insertion locations of the base model's architecture. Therefore, the fraction/percentage of added parameters may vary depending on the architecture choice of the base model. Refer to Appendix F for architecture-specific adapter locations.

Table 1: CIFAR10-to-CIFAR10C online lifelong test-time adaptation task. The numbers denote the classification error (%) obtained with the highest corruption of severity level 5. TENT-online uses domain information denoted using +. Note that FEATHER (shown in the table) **only uses 13.61% adapter parameters** added to the base model, and only these additional parameters (with BN parameters) are adapted during the test time, keeping the rest of the parameters frozen. In contrast, CoTTA requires adapting all (100%) of the parameters.

| Time | $t$ $\longrightarrow$ | | | | | | | | | | | | | | | |
|---|---|---|---|---|---|---|---|---|---|---|---|---|---|---|---|---|
| Method | Gaussian | shot | impulse | defocus | glass | motion | zoom | snow | frost | fog | brightness | contrast | elastic | pixelate | jpeg | Mean |
| Source | 72.33 | 65.71 | 72.92 | 46.94 | 54.32 | 34.75 | 42.02 | 25.07 | 41.30 | 26.01 | 9.30 | 46.69 | 26.59 | 58.45 | 30.30 | 43.51 |
| BN Adapt | 28.08 | 26.12 | 36.27 | 12.82 | 35.28 | 14.17 | 12.13 | 17.28 | 17.39 | 15.26 | 8.39 | 12.63 | 23.76 | 19.66 | 27.30 | 20.44 |
| Pseudo-label | 26.70 | 22.10 | 32.00 | 13.80 | 32.20 | 15.30 | 12.70 | 17.30 | 17.30 | 16.50 | 10.10 | 13.40 | 22.40 | 18.90 | 25.90 | 19.80 |
| TENT-online$^+$ | 24.80 | 23.52 | 33.04 | 11.93 | 31.83 | 13.71 | 10.77 | 15.90 | 16.19 | 13.67 | 7.86 | 12.05 | 21.98 | 17.29 | 24.18 | 18.58 |
| TENT-lifelong | 24.80 | **20.60** | 28.60 | 14.40 | 31.10 | 16.50 | 14.10 | 19.10 | 18.60 | 18.60 | 12.20 | 20.30 | 25.70 | 20.80 | 24.90 | 20.70 |
| CoTTA (100%) | **23.92** | 21.40 | **25.95** | **11.82** | 27.28 | 12.56 | **10.48** | 15.31 | **14.24** | 13.16 | 7.69 | **11.00** | 18.58 | 13.83 | **17.17** | **16.29** |
| FEATHER (13.61%) | 24.76 | 21.98 | 26.82 | 11.92 | 28.33 | **12.55** | 10.62 | **15.28** | 14.41 | 13.26 | 7.77 | 12.03 | 19.39 | 14.49 | 18.17 | 16.79 |

Table 2: CIFAR100-to-CIFAR100C online lifelong test-time adaptation task. The numbers denote the classification error rate (%) for the highest corruption of severity level 5. We emphasize that FEATHER (shown in the table) **only uses 6.8% adapter parameters**, added to the base model, and only these additional parameters (with BN parameters) are adapted during the test time, keeping the rest of the parameters frozen. In contrast, CoTTA requires adapting all (100%) of the parameters.

| Time | $t$ $\longrightarrow$ | | | | | | | | | | | | | | | |
|---|---|---|---|---|---|---|---|---|---|---|---|---|---|---|---|---|
| Method | Gaussian | shot | impulse | defocus | glass | motion | zoom | snow | frost | fog | brightness | contrast | elastic | pixelate | jpeg | Mean |
| Source | 73.00 | 68.01 | 39.37 | 29.32 | 54.11 | 30.81 | 28.76 | 39.49 | 45.81 | 50.30 | 29.53 | 55.10 | 37.23 | 74.69 | 41.25 | 46.45 |
| BN Adapt | 42.14 | 40.66 | 42.73 | 27.64 | 41.82 | 29.72 | 27.87 | 34.88 | 35.03 | 41.50 | 26.52 | 30.31 | 35.66 | 32.94 | 41.16 | 35.37 |
| Pseudo-label | 38.10 | 36.10 | 40.70 | 33.20 | 45.90 | 38.30 | 36.40 | 44.00 | 45.60 | 52.80 | 45.20 | 53.50 | 60.10 | 58.10 | 64.50 | 46.20 |
| TENT-lifelong | 37.20 | 35.80 | 41.70 | 37.90 | 51.20 | 48.30 | 48.50 | 58.40 | 63.70 | 71.10 | 70.40 | 82.30 | 88.00 | 88.50 | 90.40 | 60.90 |
| CoTTA (100%) | 40.09 | 37.67 | 39.77 | 26.91 | **37.82** | **28.04** | 26.26 | **32.93** | **31.72** | 40.48 | 24.72 | 26.98 | **32.33** | 28.08 | **33.46** | **32.48** |
| FEATHER (6.8%) | 40.10 | 36.66 | **38.81** | **26.68** | 38.10 | 28.56 | **25.95** | 33.81 | 32.42 | 42.12 | 24.98 | 27.32 | 34.31 | 28.60 | 35.40 | 32.92 |

Table 3: Error rate (%) results averaged over all corruption types and over 10 diverse corruption orders (highest corruption severity level 5). FEATHER adapts only a small fraction of the total number of parameters (mentioned inside brackets). CoTTA (100%) means that CoTTA requires adapting all the parameters.

| Dataset | Metric | Source | BN Adapt | TENT | CoTTA (100%) | FEATHER (10.92%) |
|---|---|---|---|---|---|---|
| ImageNet-to-ImageNetC | Error (%) | 82.35 | 72.07 | 66.52 | 63.18 | **62.64** |
| | NLL | 5.070 | 3.9956 | 3.6076 | 3.3425 | **3.3154** |
| | Brier | 0.9459 | 0.8345 | 0.8205 | 0.7681 | **0.7077** |
| ImageNet-to-ImageNet3DCC | Error (%) | 69.21 | 67.32 | 95.93 | **59.91** | 60.47 |
| | NLL | 3.9664 | 3.7163 | 19.0408 | **3.2636** | 3.3018 |
| | Brier | 0.8080 | 0.7872 | 1.8031 | **0.7270** | 0.7365 |

**CIFAR10-to-CIFAR10C:** We use pre-trained WideResNet-28 (Zagoruyko & Komodakis, 2016) as a base model for experiments on CIFAR10. For FEATHER, we add lightweight adapters with only 13.61% of the number of the base model's parameters. Table 1 reports the continual TTA error rates (Appendix E contains results on Brier score and NLL) of all the methods on CIFAR10C, where various corruptions occur continually in a sequence of mini-batches with a batch size of 200. With 86% reduction in the number of trainable/adaptable parameters, FEATHER achieves a similar average performance with a drop of 0.5% in terms of the mean error rate compared to CoTTA.

**CIFAR100-to-CIFAR100C:** For CIFAR100C, we use pre-trained ResNeXt-29 (Xie et al., 2017) as the base model. Table 2 report the error rates (Appendix E contains results on Brier score and NLL) over the sequence of corruptions. FEATHER adds only 6.8% adapter parameters to the pre-trained ResNeXt-29 for adaptation during inference. The results show that FEATHER achieves a mean error rate of 32.92% with a reduction of $\sim 93\%$ in terms of the number of trainable parameters. Adapting the entire model parameters, CoTTA achieves an improvement of only 0.34% over FEATHER in terms of average error rate. Moreover, as observed for a few of the corruptions, like shot, impulse, defocus, and zoom, FEATHER achieves a marginal improvement over CoTTA with significant savings in the parameter update cost.

**ImageNet-to-ImageNetC:** For this evaluation, we use a pre-trained ResNet-50 as the base model. In this setting, prior works (Wang et al., 2022) report the lifelong TTA performance over 10 random sequences of the 15 corruptions. To provide a fair comparison, we experiment with FEATHER, considering the same continual setting where the performances are validated for 10 random sequences of corruptions. Table 3 shows the performance over 10 different runs. We observe that with only 10.92% of added trainable adapter parameters, FEATHER achieves an improvement in terms of error rate over CoTTA (Wang et al., 2022) (from 63.18% to 62.64%). This highlights that the parameter update cost can be significantly reduced for existing approaches with no performance drop.

**ImageNet-to-ImageNet3DCC:** For this evaluation, we use the same architecture as that of ImageNetC experiments with a pre-trained ResNet-50 and the same number of added adaptable parameters (10.92%). Table 3 highlights the performance over 10 random orders of corruptions. We observe that with only 10.92% of added adaptable parameters, FEATHER achieves a comparable average error rate of 60.47% compared to an average error rate of 59.91% for CoTTA (with all parameters), with only 0.56% performance drop.

Overall, our detailed results in four benchmarks highlight that FEATHER achieves a comparable performance with huge efficiency in the number of trainable parameters. Fig. 2 highlights the comparable performance achieved with significant efficiency using FEATHER over CoTTA. In Table 4, we report the comparison with other existing SOTA approaches in a lifelong TTA setting. Refer to Appendix A for more details about the hyperparameters.

Table 4: The table shows the comparison of FEATHER with other existing TTA methods in terms of the error rate (%) on CIFAR10-to-CIFAR10C and ImageNet-to-ImageNetC datasets. Note that FEATHER uses the learning objective and TTA scheme proposed by CoTTA, and all the comparisons are made in a lifelong setting.

| Method | CIFAR10C | ImageNetC | Source Free |
|---|---|---|---|
| Source | 43.51 | 82.35 | ✓ |
| BN Stats | 20.44 | 72.07 | ✓ |
| TENT | 20.7 | 66.52 | ✓ |
| EATA | 18.6 | 63.8 | ✗ |
| NOTE | 20.2 | - | ✓ |
| ECoTTA | 16.8 | 63.4 | ✗ |
| CoTTA | **16.29** | 63.18 | ✓ |
| FEATHER | 16.79 | **62.64** | ✓ |

## 6 DISCUSSION

**Flexibility in Parameter Efficiency:** The overall objective of the test-time adaptation methods is to increase the usage of existing methods in the real-world changing environment over time, making the models more robust towards domain shifts when deployed in the wild.

However, it is imperative that a proposed method does not compromise upon the predictive performance. FEATHER provides flexibility in choosing the desired number of additional adapter parameters for a task. To validate if the same performance can be achieved by adding more adapter parameters ($\omega$), we experiment with the FEATHER setting, where we increase the trainable number of parameters by adding more adapters to the base model. We experiment with multiple settings where we add different number of parameters. Table 5 highlights the performance comparison along with the parameter comparison in

Table 5: Error rate (%) on CIFAR100C over different percentages of added parameters in FEATHER. Param. % in the bracket in the first two columns indicates a comparison with the base model, for e.g., 49M (7.16% of 6.90M) and 7.37M (106.80% of 6.9M). We observe that adding a similar number of trainable parameters as adapters (101.29%, last row) improves the performance over CoTTA by a small margin, and even with a much smaller number of trainable params., FEATHER achieves comparable performance.

| Method | Train. Params. | Total Params. | Train. % | Error |
|---|---|---|---|---|
| **CoTTA** | 6.90M | 6.90M | 100.00% | 32.5 |
| **FEATHER** | 0.49M (7.16%) | 7.37M (106.80%) | **6.71%** | 32.92 |
|  | 2.35M (34.12%) | 9.23M (133.76%) | 25.51% | 32.79 |
|  | 3.94M (57.14%) | 10.82M (156.78%) | 36.45% | 32.65 |
|  | 6.99M (101.29%) | 13.89M (201.29%) | 50.32% | **32.31** |

detail. As observed from the results, increasing the number of adapter parameters does help boost the performance and making the trainable parameters 57.14% of the base model achieves 32.65% mean error rate, which is very close to CoTTA which requires retraining all (i.e., 100%) the parameters. Moreover, we also observe that adding a similar number of trainable parameters using FEATHER (101.29% of the base model) helps achieve marginal performance improvement over the CoTTA baseline (32.5% to 32.31% mean error rate).

**Reset Cost:** While deploying models in the field, a long run of domain adaptation causes parameter drift and performance degradation. For instance, the occurrence of highly shifted domains during testing may lead to a significant parameter drift, resulting in the loss of all source knowledge. Therefore, existing approaches (Wang et al., 2021; Song et al., 2023) typically maintain a copy of the source model for resetting the model parameters back to the source parameters. In FEATHER,

Table 6: The table compares FEATHER applied over TENT and CoTTA, highlighting the orthogonality of the proposed generic framework. Results for the CIFAR100-to-CIFAR100C benchmark depict the parameter efficiency obtained (**93.2% fewer trainable parameters** compared to CoTTA on the base model)

| Time | $t \longrightarrow$ | | | | | | | | | | | | | | | |
|---|---|---|---|---|---|---|---|---|---|---|---|---|---|---|---|---|
| Method | Gaussian | shot | impulse | defocus | glass | motion | zoom | snow | frost | fog | brightness | contrast | elastic | pixelate | jpeg | Mean |
| **TENT-lifelong** | **37.20** | **35.80** | 41.70 | 37.90 | 51.20 | 48.30 | 48.50 | 58.40 | 63.70 | 71.10 | 70.40 | 82.30 | 88.00 | 88.50 | 90.40 | 60.90 |
| **+ FEATHER (6.8% Params)** | 41.67 | 39.40 | **41.35** | **26.96** | **40.11** | **28.87** | **26.91** | **33.87** | **33.80** | **39.94** | **26.27** | **29.55** | **34.50** | **32.02** | **40.10** | **34.35** |
| **EATA-lifelong** | **41.83** | **40.27** | **42.56** | **27.56** | **41.54** | **29.54** | **27.70** | **34.69** | **34.71** | **41.24** | **26.42** | **30.20** | **35.58** | **32.73** | **40.95** | **35.17** |
| **+ FEATHER (6.8% Params)** | 41.21 | 38.96 | 41.24 | 26.97 | 41.07 | 29.26 | 27.13 | 33.84 | 34.30 | 39.94 | 26.04 | 30.14 | 34.61 | 31.67 | 39.74 | 34.41 |
| **CoTTA (100% Params)** | **40.09** | 37.67 | 39.77 | 26.91 | **37.82** | 28.04 | 26.26 | 32.93 | 31.72 | 40.48 | 24.72 | 26.98 | 32.33 | 28.08 | 33.46 | **32.48** |
| **+ FEATHER (6.8% Params)** | 40.10 | **36.66** | **38.81** | **26.68** | 38.10 | 28.56 | **25.95** | 33.81 | 32.42 | 42.12 | 24.98 | 27.32 | 34.31 | 28.60 | 35.40 | 32.92 |

since we only update the newly added parameters, keeping the source parameters untouched, the models adapted using FEATHER can be easily reset to source without requiring a copy of the source model. Therefore FEATHER, by its design itself, helps reduce the *reset cost* for a model in terms of memory required for deploying the model in the wild.

**Orthogonality with existing TTA approaches:** As FEATHER emphasizes parameter efficiency, the proposed method and the learning objective $\mathcal{L}_{\mathcal{U}}(f_\omega())$ defined with respect to adapter parameters ($\omega$) is generic and can be any unsupervised test-time adaptation loss, making it orthogonal to existing approaches. To validate the orthogonality with existing LTTA approaches, we perform another set of experiments where we combine FEATHER with the learning objectives proposed by TENT, EATA, and CoTTA. Table 6 reports the results for FEATHER using the learning objective of TENT, EATA, and CoTTA. Note that TENT and EATA propose adapting only BN parameters (0.37%) during TTA, whereas CoTTA adapts the entire network weights (100%).

Table 7 depicts the decrease in inference/TTA time and memory budget requirement, along with the error rate comparison between various architecture settings and different lifelong TTA methods for the CIFAR100-to-CIFAR100C benchmark. Even though updating *BN Params* adapts a minimal number of parameters, FEATHER

Table 7: Time and memory budget requirements for CIFAR100-to-CIFAR100C upon varying number of trainable parameters, along with the error rate (%).

| | Adapt. Params | Mem. (MB) | TTA Time (secs) \| Error (%) | | |
|---|---|---|---|---|---|
| | | | CoTTA | TENT | EATA |
| BN Params | 0.37% | 2890.56 | 110.80 \| 34.70 | 18.60 \| 60.90 | 28.06 \| 35.17 |
| All Params | 100% | 5693.27 | 149.70 \| 32.48 | 28.53 \| 33.64 | 33.80 \| 33.89 |
| FEATHER | 6.80% | 3311.87 | 136.14 \| 32.92 | 21.20 \| 32.92 | 28.60 \| 34.41 |

with only 6.80% adaptable parameters outperforms *BN Params* significantly and performs comparably with *All Params* version, consistently across CoTTA, TENT, and EATA. Thus, FEATHER provides an advantage of adapting a minuscule percentage of parameters to achieve similar/better performance based on the available memory/time budget, making it more practical and flexible for real-life deployment of the TTA models.

## 7 CONCLUSION

In this work, we propose a generic framework for making adaptation efficient during test time and introduce FEATHER: liFelong tEst-time Adaptation wiTH lightwEight adapteRs. FEATHER uses efficient adapters, which can be trained at test-time (using unlabeled test inputs) to improve the performance under domain shifts. FEATHER offers two key advantages: making the adaptation parameter efficient and keeping the source knowledge intact. With the proposed initialization scheme, FEATHER also removes the dependency on source dataset at the adaptation time (required by other recent methods), making the proposed adapters compatible with the full-test-time adaptation setting. FEATHER requires substantially (**85% to 94%**) fewer trainable parameters to achieve better or similar performance compared to existing TTA state-of-the-art methods, resulting in faster adaptation and inference during test-time. The proposed adapters and initialization scheme will help provide parameter control to the test-time adaptation approaches and make them more efficient for real-world use cases. We conclude by mentioning a few avenues of possible future work: (1) making the parameter addition dynamic (adding parameters on the fly) based on the observed domain shifts (currently, they are fixed); (2) using adapters with low-rank structures (Hu et al., 2021) to further improve the parameter efficiency; and (3) designing variants of FEATHER for other architectures (Transformers (Kojima et al., 2022), Graph Neural Networks (Jin et al., 2023), etc.).

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

APPENDIX

## A HYPERPARAMETER SETTING

We use PyTorch (Paszke et al., 2019) to develop and train our architecture and RobustBench Croce et al. (2021) for the various pre-trained architectures used in the experiments. We use the existing set of pre-trained models (base models) widely used for TTA experiments. We run all the experiments over the NVIDIA A40 GPU. For hyperparameters, we make use of the available set of hyperparameters proposed by the TTA approaches; for example, for using the TTA mechanism proposed by CoTTA, we use hyperparameters provided by CoTTA. For a fair comparison, we use the same optimizers (Adam (Kingma & Ba, 2014) and SGD) as reported by previous TTA baselines. As we decrease the number of parameters by a significant margin, we tune the learning rate for various settings.

For CIFAR10-to-CIFAR10C experiments, we use Adam optimizer (Kingma & Ba, 2014) with a learning rate of $0.00125$, and $\beta = 0.9$ with no weight decay. We follow CoTTA for the mean teacher parameter and update the weights of the teacher model by exponential moving average using the student model weights using $\alpha = 0.999$, but without any resetting, i.e., reset rate of $0\%$.

For CIFAR100-to-CIFAR100C experiments, we use a learning rate of $0.0015$ for Adam optimizer with $\beta = 0.9$ and no weight decay. For the mean teacher parameter, we follow CoTTA and update the weights of the teacher model by exponential moving average using the student model weights using $\alpha = 0.999$ with a reset rate of $1\%$.

In the ImageNet-to-ImageNetC experiments, we use stochastic gradient descent (SGD) optimizer with a learning rate of $0.04$, momentum of $0.9$, and no weight decay. We follow CoTTA for the mean teacher weight parameter and update teacher model weights by exponential moving average using student model weights with $\alpha = 0.999$, and a reset rate of $0.1\%$.

The ImageNet-to-ImageNet3DCC experiments use SGD optimizer with a learning rate of $0.03$, a momentum of $0.9$, with no weight decay. We follow CoTTA for the mean teacher weight parameter. The teacher model weights are updated by exponential moving average, utilizing the student model weights with $\alpha = 0.999$ and a reset rate of $0.1\%$.

## B BENCHMARK DATASETS

Multiple corruptions are introduced to the standard CIFAR10 and CIFAR100 Krizhevsky (2009) datasets to get CIFAR10C and CIFAR100C datasets are corrupted versions, respectively. Both ImageNetC Hendrycks & Dietterich (2019) and ImageNet3DCC Kar et al. (2022) are corrupted versions of the standard ImageNet Deng et al. (2009) dataset.

The CIFAR10C and CIFAR100C datasets both consist of 10,000 images for every corruption type, resulting in a total of 150,000 images for each dataset. The ImageNetC dataset consists of 50,000 images for each corruption class. The CIFAR10C, CIFAR100C, and ImageNet-C datasets comprise a total of 15 distinct types of corruptions, with an additional 4 types designated for validation purposes. Every corruption consists of five distinct levels of severity. The different kinds of corruption, accompanied by concise explanations, are outlined below:

1. Gaussian noise: frequently observed in situations characterized by low illumination levels
2. Shot noise: electrical noise that arises from the discrete character of light
3. Impulse noise: color counterpart of salt-and-pepper noise and can potentially occur owing to bit errors
4. Defocus blur: occurs when an image is captured with an improper focus
5. Frosted glass blur: occurs when an image is seen through a window having frosted glass
6. Motion blur: a phenomenon that arises when a camera undergoes rapid movement
7. Zoom blur: occurs when a camera rapidly moves toward an object
8. Snow: a form of precipitation that visibly obscures the object of interest

9. Frost: happens when ice crystals stick to windows

10. Fog: the presence of fog in the environment causes objects to be obscured from view; this effect is generated using the diamond-square algorithm

11. Brightness: subject to change in accordance with the intensity of sunlight

12. Contrast: influenced by the lighting conditions and the color of the photographed object

13. Elastic transformations: stretching or contracting of small image portions

14. Pixelation: occurs when a low-resolution image is upsampled

15. JPEG: lossy image compression method that leads to the formation of compression artifacts

The ImageNet 3D Common Corruptions (ImageNet3DCC) dataset, as proposed in a recent work by Kar et al. (2022), utilizes the scene geometry for transformations, leading to the generation of corruptions more closely resembling real-world scenarios. The Imagenet3DCC dataset consists of 50,000 images for every form of corruption included within the dataset. It comprises a total of 12 different kinds of corruption, each characterized by 5 degrees of severity. The instances of corruption can be categorized as follows:

1. Near focus: altering the focus region to the nearby section of the scene in a random manner

2. Far focus: introduce random alterations in the focus to encompass the far portion of the scene

3. Bit error: attributed to the presence of imperfections in the video transmission channel

4. Color quantization: reduces the bit depth of an RGB image

5. Flash: occurs when a light source is placed in close proximity to the camera

6. Fog 3D: produced by utilizing a conventional optical model for fog

7. H.265 ABR: H.265 codec in conjunction with the Average Bit Rate control mode for compression purposes

8. H.265 CRF: H.265 codec for compression purposes, specifically employing the Constant Rate Factor (CRF) control mode

9. ISO noise: refers to the presence of noise in an image, which follows a Poisson-Gaussian distribution

10. Low-light: simulated by lowering pixel intensities and addition of Poisson-Gaussian distributed noise

11. XY-motion blur: refers to the blur when the primary camera is in motion along the XY-plane of the picture

12. Z-motion blur: occurs when the primary camera is moving along the Z-axis of the image

The purpose of developing these datasets is to provide standardized benchmarks for evaluating the robustness of classification models.

## C  EVALUATION METRICS

For a given dataset, assume $D = \{x_n, y_n\}_{n=1}^N$, with $y_n$ to be the true label (in one-hot representation, i.e., $y_{ni} = 1$ if $i$ is the true class label, else $y_{ni} = 0$) of $x_n$, and $y'_n$ to be the prediction by the model.

### C.1  ERROR

The definition of average error rate is as follows:

$$\text{Error} = \frac{1}{N} \sum_{n=1}^{N} \mathbb{I}(y'_n \neq y_n). \tag{1}$$

Here $\mathbb{I}()$ denotes the indicator function.

## C.2 BRIER SCORE

The average Brier score Brier (1950) is given by the following:

$$\text{Brier score} = \frac{1}{N} \sum_{n=1}^{N} \sum_{i=1}^{D} (y'_{ni} - y_{ni})^2. \tag{2}$$

## C.3 NEGATIVE LOG-LIKELIHOOD

We define average negative log-likelihood (NLL) as:

$$\text{NLL} = -\frac{1}{N} \sum_{n=1}^{N} \sum_{i=1}^{D} (y_{ni} \log y'_{ni}). \tag{3}$$

## D INITIALIZATION SCHEME ADAPTER PARAMETERS

We provide a pictorial depiction of the proposed initialization scheme to make the adapters behave as an identity function in the beginning. This ensures that source domain knowledge is intact and, thus, no warm-up using the source data is required, unlike approaches such as Niu et al. (2022); Song et al. (2023).

## E ADDITIONAL RESULTS

For a fair comparison with existing TTA methods, we report all the metrics results corresponding to Table 1 and Table 2 in Table 8 and Table 9, respectively. Overall, we observe that similar performance can be achieved with a significant reduction in trainable parameters.

**Comparison with other SOTA methods:** The main focus of this work is to design parameter-efficient adapters that are well-suited for lifelong test-time adaptation (TTA). All comparisons conducted in our work are based on the mechanism proposed by CoTTA. In addition, for completeness, we also report a comparative analysis of the performance of our approach with some other recently proposed lifelong TTA approaches.

Table 4 provides a comparison with other existing methods. Since different methods show results on various architectures, we typically use the standard architectures and report the numbers from the paper corresponding to the same architecture and dataset. Note that another recent work EcoTTA (Song et al., 2023) proposes to include meta-network modules to the base model for reducing the activation and memory cost in TTA methods. Adding more parameters in EcoTTA still requires a warm-up phase, making it dependent on the availability of the source dataset. In contrast, our approach FEATHER removes this dependency by proposing adapter designs compatible with identity transformation for initialization, making it generic for all TTA methods.

**Orthogonality with existing TTA approaches:** TENT and EATA only make use of BN parameters in their proposed approach. When compared to the BN parameters adaptable version of both TENT and EATA, TENT+FEATHER and EATA+FEATHER both achieve significant performance improvement; we speculate the primary reason to be the usage of more parameter space for adaptation without losing the source model weights. Hence, to evaluate the dependence throughout the adaptable parameter space, we create an additional setting in which we update the complete model parameters (100% trainable parameters) and report the findings. Similarly, for comparison with CoTTA, we add an additional setting of adapting only BN parameters. Experimental results in Table 10 show that adapting the entire model parameters does help boost the performance by a significant margin; however, it loses the proxy for source knowledge as all the parameters are now updated. We discover that FEATHER can achieve a similar performance improvement with a modest fraction of extra adapter parameters, allowing us to maintain the performance boost with great parameter efficiency without any loss in original model parameters following an update.

In terms of error rate, Table 10 demonstrates that CoTTA with only FEATHER adapters being adaptable has an error rate of 34.70%, surpassing CoTTA with only BN params being adaptable, which has an error rate of 32.48%.

Table 8: The table shows all the metrics results obtained for the CIFAR10-to-CIFAR10C online lifelong test-time adaptation task for the highest corruption of severity level 5 corresponding to the results obtained for `FEATHER` reported in Table 1. `FEATHER` here only uses 13.61% adapter parameters compared to the base model. Note that `FEATHER` here uses the learning objective and TTA scheme proposed by CoTTA.

| Method | Metric | Gaussian | shot | impulse | defocus | glass | motion | zoom | snow | frost | fog | brightness | contrast | elastic | pixelate | jpeg | Mean |
|---|---|---|---|---|---|---|---|---|---|---|---|---|---|---|---|---|---|
| Source | Error % | 72.33 | 65.71 | 72.92 | 46.94 | 54.32 | 34.75 | 42.02 | 25.07 | 41.30 | 26.01 | 9.30 | 46.69 | 26.59 | 58.45 | 30.30 | 43.51 |
|  | Brier | 1.29 | 1.16 | 1.21 | 0.79 | 0.93 | 0.59 | 0.71 | 0.42 | 0.72 | 0.44 | 0.15 | 0.77 | 0.44 | 1.02 | 0.50 | 0.74 |
|  | NLL | 6.46 | 5.61 | 5.47 | 2.74 | 3.84 | 2.09 | 2.51 | 1.51 | 3.15 | 1.53 | 0.48 | 2.69 | 1.38 | 4.67 | 1.65 | 3.05 |
| BN | Error % | 28.08 | 26.12 | 36.27 | 12.82 | 35.28 | 14.17 | 12.13 | 17.28 | 17.39 | 15.26 | 8.39 | 12.63 | 23.76 | 19.66 | 27.30 | 20.44 |
|  | Brier | 0.46 | 0.43 | 0.59 | 0.20 | 0.57 | 0.23 | 0.19 | 0.28 | 0.28 | 0.24 | 0.13 | 0.20 | 0.38 | 0.32 | 0.45 | 0.33 |
|  | NLL | 1.46 | 1.32 | 1.90 | 0.57 | 1.76 | 0.64 | 0.54 | 0.82 | 0.82 | 0.71 | 0.36 | 0.57 | 1.14 | 0.92 | 1.38 | 0.99 |
| TENT | Error % | 24.80 | 20.48 | 28.49 | 14.84 | 31.78 | 16.97 | 16.66 | 21.97 | 20.97 | 20.92 | 14.76 | 19.91 | 27.56 | 23.89 | 31.01 | 22.33 |
|  | Brier | 0.42 | 0.35 | 0.50 | 0.26 | 0.56 | 0.30 | 0.30 | 0.40 | 0.39 | 0.38 | 0.27 | 0.37 | 0.51 | 0.44 | 0.58 | 0.40 |
|  | NLL | 1.41 | 1.33 | 2.10 | 0.57 | 2.61 | 1.52 | 1.65 | 2.34 | 2.43 | 2.42 | 1.76 | 2.48 | 3.21 | 2.97 | 4.17 | 2.23 |
| CoTTA | Error % | 23.92 | 21.40 | 25.95 | 11.82 | 27.28 | 12.56 | 10.48 | 15.31 | 14.24 | 13.16 | 7.69 | 11.00 | 18.58 | 13.83 | 17.17 | 16.29 |
|  | Brier | 0.36 | 0.33 | 0.38 | 0.18 | 0.40 | 0.19 | 0.16 | 0.23 | 0.21 | 0.20 | 0.11 | 0.16 | 0.27 | 0.20 | 0.25 | 0.24 |
|  | NLL | 0.92 | 0.85 | 0.88 | 0.43 | 0.93 | 0.46 | 0.37 | 0.55 | 0.50 | 0.46 | 0.26 | 0.36 | 0.60 | 0.45 | 0.56 | 0.57 |
| FEATHER | Error % | 24.76 | 21.98 | 26.82 | 11.92 | 28.33 | 12.55 | 10.62 | 15.28 | 14.41 | 13.26 | 7.77 | 12.03 | 19.39 | 14.49 | 18.17 | 16.79 |
|  | Brier | 0.38 | 0.34 | 0.39 | 0.18 | 0.42 | 0.19 | 0.16 | 0.23 | 0.21 | 0.20 | 0.11 | 0.17 | 0.28 | 0.21 | 0.27 | 0.25 |
|  | NLL | 1.02 | 0.92 | 0.92 | 0.44 | 0.98 | 0.45 | 0.37 | 0.54 | 0.50 | 0.45 | 0.24 | 0.39 | 0.62 | 0.47 | 0.60 | 0.59 |

Table 9: The table shows all the metrics results obtained for the CIFAR100-to-CIFAR100C online lifelong test-time adaptation task for the highest corruption of severity level 5 corresponding to the results obtained for `FEATHER` reported in Table 2. `FEATHER` here only uses 6.8% adapter parameters compared to the base model. Note that `FEATHER` here, uses the learning objective and TTA scheme proposed by CoTTA.

| Method | Metric | Gaussian | shot | impulse | defocus | glass | motion | zoom | snow | frost | fog | brightness | contrast | elastic | pixelate | jpeg | Mean |
|---|---|---|---|---|---|---|---|---|---|---|---|---|---|---|---|---|---|
| Source | Error % | 73.00 | 68.01 | 39.37 | 29.32 | 54.11 | 30.81 | 28.76 | 39.49 | 45.81 | 50.30 | 29.53 | 55.10 | 37.23 | 74.69 | 41.25 | 46.45 |
|  | Brier | 1.11 | 1.04 | 0.58 | 0.41 | 0.79 | 0.43 | 0.40 | 0.53 | 0.64 | 0.71 | 0.41 | 0.75 | 0.51 | 1.12 | 0.56 | 0.67 |
|  | NLL | 5.59 | 4.89 | 2.00 | 1.19 | 2.86 | 1.26 | 1.16 | 1.63 | 2.12 | 2.34 | 1.16 | 2.52 | 1.50 | 5.39 | 1.74 | 2.49 |
| BN | Error % | 42.14 | 40.66 | 42.73 | 27.64 | 41.82 | 29.72 | 27.87 | 34.88 | 35.03 | 41.50 | 26.52 | 30.31 | 35.66 | 32.94 | 41.16 | 35.37 |
|  | Brier | 0.55 | 0.54 | 0.56 | 0.37 | 0.55 | 0.40 | 0.38 | 0.47 | 0.46 | 0.55 | 0.36 | 0.40 | 0.48 | 0.44 | 0.54 | 0.47 |
|  | NLL | 1.69 | 1.62 | 1.71 | 1.06 | 1.64 | 1.13 | 1.06 | 1.38 | 1.37 | 1.66 | 1.01 | 1.17 | 1.40 | 1.29 | 1.66 | 1.39 |
| TENT | Error % | 37.16 | 35.61 | 41.82 | 37.54 | 51.19 | 48.48 | 49.15 | 58.83 | 62.85 | 71.65 | 70.76 | 82.91 | 88.00 | 91.14 | 94.63 | 61.45 |
|  | Brier | 0.51 | 0.52 | 0.63 | 0.60 | 0.82 | 0.82 | 0.8585 | 1.03 | 1.13 | 1.31 | 1.32 | 1.60 | 1.68 | 1.77 | 1.85 | 1.10 |
|  | NLL | 1.49 | 1.58 | 2.14 | 2.12 | 3.28 | 3.66 | 4.17 | 5.46 | 6.71 | 8.53 | 9.04 | 14.43 | 14.17 | 16.21 | 17.66 | 7.37 |
| CoTTA | Error % | 40.09 | 37.67 | 39.77 | 26.91 | 37.82 | 28.04 | 26.26 | 32.93 | 31.72 | 40.48 | 24.72 | 26.98 | 32.33 | 28.08 | 33.46 | 32.48 |
|  | Brier | 0.53 | 0.51 | 0.53 | 0.37 | 0.50 | 0.38 | 0.36 | 0.44 | 0.43 | 0.53 | 0.35 | 0.37 | 0.44 | 0.39 | 0.45 | 0.44 |
|  | NLL | 1.60 | 1.50 | 1.58 | 1.04 | 1.47 | 1.09 | 1.02 | 1.29 | 1.24 | 1.59 | 0.96 | 1.05 | 1.25 | 1.10 | 1.30 | 1.27 |
| FEATHER | Error % | 40.10 | 36.66 | 38.81 | 26.68 | 38.10 | 28.56 | 25.95 | 33.81 | 32.42 | 42.12 | 24.98 | 27.32 | 34.31 | 28.60 | 35.40 | 32.92 |
|  | Brier | 0.53 | 0.5 | 0.52 | 0.37 | 0.51 | 0.39 | 0.36 | 0.46 | 0.44 | 0.55 | 0.35 | 0.38 | 0.46 | 0.39 | 0.47 | 0.45 |
|  | NLL | 1.6 | 1.46 | 1.51 | 1 | 1.46 | 1.07 | 0.97 | 1.3 | 1.22 | 1.65 | 0.93 | 1.03 | 1.28 | 1.07 | 1.34 | 1.26 |

**Flexibility in Parameter Efficiency:** We report the exact number of parameters for Table 5 in the Table 11. This illustrates the flexibility of `FEATHER` to choose the number of parameters depending on the memory and computational budget.

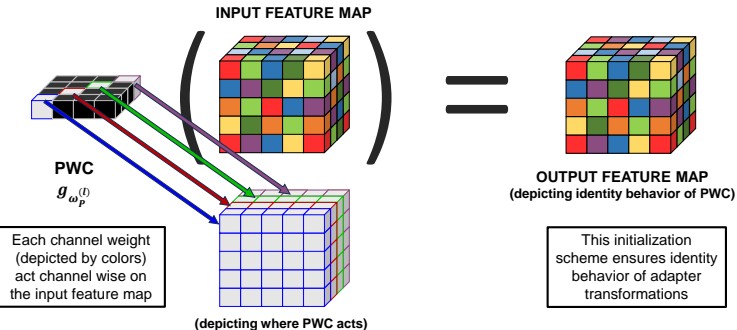

Figure 3: The figure highlights the proposed initialization scheme for the added adapter parameters. Specifically, the Point Wise Convolution (PWC), when initialized with diagonal elements (highlighted in white), acts channelwise on the input feature map, making the interaction between the channels zero and projecting the same feature space to act as the output feature map.

Table 10: The table compares FEATHER applied over TENT and CoTTA, highlighting the orthogonality of the proposed generic framework. Results for the CIFAR100-to-CIFAR100C benchmark depict the parameter efficiency obtained (**93.2% fewer parameters**) with a meager performance drop (0.71% for TENT (100% params), 0.52% for EATA (100% params), and 0.44% for CoTTA).

| Time | $t$ ⟶ | | | | | | | | | | | | | | | |
|---|---|---|---|---|---|---|---|---|---|---|---|---|---|---|---|---|
| **Method** | **Gaussian** | **shot** | **impulse** | **defocus** | **glass** | **motion** | **zoom** | **snow** | **frost** | **fog** | **brightness** | **contrast** | **elastic** | **pixelate** | **jpeg** | **Mean** |
| **TENT-lifelong** | **37.20** | 35.80 | 41.70 | 37.90 | 51.20 | 48.30 | 48.50 | 58.40 | 63.70 | 71.10 | 70.40 | 82.30 | 88.00 | 88.50 | 90.40 | 60.90 |
| **TENT-lifelong (100% Params)** | 40.27 | **35.68** | **37.28** | 26.27 | **37.81** | 28.98 | 26.97 | **33.39** | **32.52** | **39.21** | 27.33 | 32.39 | 34.87 | 32.03 | **39.65** | **33.64** |
| **+ FEATHER (6.8% Params)** | 41.67 | 39.40 | 41.35 | 26.96 | 40.11 | **28.87** | **26.91** | 33.87 | 33.80 | 39.94 | **26.27** | **29.55** | **34.50** | **32.02** | 40.10 | 34.35 |
| **EATA-lifelong** | 41.83 | 40.27 | 42.56 | 27.56 | 41.54 | 29.54 | 27.70 | 34.69 | 34.71 | 41.24 | 26.42 | 30.2 | 35.58 | 32.73 | 40.95 | 35.17 |
| **EATA-lifelong (100% Params)** | 41.50 | **38.81** | **41.07** | 26.82 | **39.90** | 28.90 | 26.83 | **33.56** | **33.11** | **39.60** | 25.38 | **29.00** | **34.15** | 30.79 | **38.96** | **33.89** |
| **+ FEATHER (6.8% Params)** | **41.21** | 38.96 | 41.24 | 26.97 | 41.07 | 29.26 | 27.13 | 33.84 | 34.3 | 39.94 | 26.04 | 30.14 | 34.61 | 31.67 | 39.74 | 34.41 |
| **CoTTA (BN Params)** | 40.33 | 38.3 | 40.16 | 27.67 | 39.99 | 29.76 | 27.88 | 35.51 | 34.68 | 43.4 | 26.58 | 30.52 | 35.98 | 32.05 | 37.71 | 34.70 |
| **CoTTA (100% Params)** | **40.09** | 37.67 | 39.77 | 26.91 | **37.82** | 28.04 | 26.26 | **32.93** | **31.72** | **40.48** | 24.72 | **26.98** | **32.33** | 28.08 | 33.46 | **32.48** |
| **+ FEATHER (6.8% Params)** | 40.10 | **36.66** | **38.81** | 26.68 | 38.10 | 28.56 | **25.95** | 33.81 | 32.42 | 42.12 | 24.98 | 27.32 | 34.31 | 28.60 | 35.40 | 32.92 |

Table 11: Error rate (%) on CIFAR100C over different percentages of added parameters in FEATHER. The (Parameter %) in the bracket in the first two columns indicates a comparison with the base model. For example, 494208 (7.16% of 6900132) and 7369124 (106.80% of 6900132). We observe that adding a similar number of trainable parameters as adapters (101.29%, last row) improves the performance over CoTTA by a small margin, and even with a much smaller number of trainable parameters, FEATHER achieves comparable performance.

| Method | Trainable params | Total Params | Trainable % | CIFAR100C |
|---|---|---|---|---|
| CoTTA | 6900132 | 6900132 | 100.00% | 32.5 |
| FEATHER | 494208 (7.16%) | 7369124 (106.80%) | **6.71%** | 32.92 |
| | 2354816 (34.12%) | 9229732 (133.76%) | 25.51% | 32.79 |
| | 3943040 (57.14%) | 10817956 (156.78%) | 36.45% | 32.65 |
| | 6988800 (101.29%) | 13888932 (201.29%) | 50.32% | **32.31** |

## F   ARCHITECTURE DETAILS ALONG WITH FEATHER ADAPTERS

In this section, we report the architecture-specific details used for adapter parameters.  For CIFAR100-to-CIFAR100C experiments, we modify the widely used ResNeXt-29 architecture taken from RobustBench Croce et al. (2021) and added adapters in between referred to as ConvAdapt layers. For ImageNet-to-ImageNetC, we modify the ResNet-50 architecture and add adapter layers in between. The added adapter layers are kept in bold.

ResNeXt-29 with adapters for CIFAR100-to-CIFAR100C

```
Hendrycks2020AugMixResNeXtNetAdpt(
        (conv_1_3x3): Conv2d(3, 64, kernel_size=(3, 3), stride=(1, 1), padding=(1, 1), bias=False)
  (bn_1): BatchNorm2d(64, eps=1e-05, momentum=0.1, affine=True, track_running_stats=False)
  (stage_1): Sequential(
    (0): ResNeXtBottleneck(
      (conv_reduce): Conv2d(64, 128, kernel_size=(1, 1), stride=(1, 1), bias=False)
      (bn_reduce): BatchNorm2d(128, eps=1e-05, momentum=0.1, affine=True, track_running_stats=False)
      (conv_conv): Conv2d(128, 128, kernel_size=(3, 3), stride=(1, 1), padding=(1, 1), groups=4, bias=False)
      (bn): BatchNorm2d(128, eps=1e-05, momentum=0.1, affine=True, track_running_stats=False)
      (conv_expand): Conv2d(128, 256, kernel_size=(1, 1), stride=(1, 1), bias=False)
      (bn_expand): BatchNorm2d(256, eps=1e-05, momentum=0.1, affine=True, track_running_stats=False)
      (downsample): Sequential(
        (0): Conv2d(64, 256, kernel_size=(1, 1), stride=(1, 1), bias=False)
        (1): BatchNorm2d(256, eps=1e-05, momentum=0.1, affine=True, track_running_stats=False)
      )
    )
    (1): ResNeXtBottleneck(
      (conv_reduce): Conv2d(256, 128, kernel_size=(1, 1), stride=(1, 1), bias=False)
      (bn_reduce): BatchNorm2d(128, eps=1e-05, momentum=0.1, affine=True, track_running_stats=False)
      (conv_conv): Conv2d(128, 128, kernel_size=(3, 3), stride=(1, 1), padding=(1, 1), groups=4, bias=False)
      (bn): BatchNorm2d(128, eps=1e-05, momentum=0.1, affine=True, track_running_stats=False)
      (conv_expand): Conv2d(128, 256, kernel_size=(1, 1), stride=(1, 1), bias=False)
      (bn_expand): BatchNorm2d(256, eps=1e-05, momentum=0.1, affine=True, track_running_stats=False)
    )
    (2): ResNeXtBottleneck(
      (conv_reduce): Conv2d(256, 128, kernel_size=(1, 1), stride=(1, 1), bias=False)
      (bn_reduce): BatchNorm2d(128, eps=1e-05, momentum=0.1, affine=True, track_running_stats=False)
      (conv_conv): Conv2d(128, 128, kernel_size=(3, 3), stride=(1, 1), padding=(1, 1), groups=4, bias=False)
      (bn): BatchNorm2d(128, eps=1e-05, momentum=0.1, affine=True, track_running_stats=False)
      (conv_expand): Conv2d(128, 256, kernel_size=(1, 1), stride=(1, 1), bias=False)
      (bn_expand): BatchNorm2d(256, eps=1e-05, momentum=0.1, affine=True, track_running_stats=False)
    )
  )
  (stage_2): Sequential(
    (0): ResNeXtBottleneck(
      (conv_reduce): Conv2d(256, 256, kernel_size=(1, 1), stride=(1, 1), bias=False)
      (bn_reduce): BatchNorm2d(256, eps=1e-05, momentum=0.1, affine=True, track_running_stats=False)
      (conv_conv): Conv2d(256, 256, kernel_size=(3, 3), stride=(2, 2), padding=(1, 1), groups=4, bias=False)
      (bn): BatchNorm2d(256, eps=1e-05, momentum=0.1, affine=True, track_running_stats=False)
      (conv_expand): Conv2d(256, 512, kernel_size=(1, 1), stride=(1, 1), bias=False)
      (bn_expand): BatchNorm2d(512, eps=1e-05, momentum=0.1, affine=True, track_running_stats=False)
      (downsample): Sequential(
        (0): Conv2d(256, 512, kernel_size=(1, 1), stride=(2, 2), bias=False)
        (1): BatchNorm2d(512, eps=1e-05, momentum=0.1, affine=True, track_running_stats=False)
      )
    )
    (1): ResNeXtBottleneckAdpt(
      (conv_reduce): Conv2d(512, 256, kernel_size=(1, 1), stride=(1, 1), bias=False)
      (bn_reduce): BatchNorm2d(256, eps=1e-05, momentum=0.1, affine=True, track_running_stats=False)
      (lhc1): ConvAdapt(
        (gwc): Conv2d(256, 256, kernel_size=(3, 3), stride=(1, 1), padding=(1, 1), groups=32)
        (pwc): Conv2d(256, 256, kernel_size=(1, 1), stride=(1, 1))
      )
      (conv_conv): Conv2d(256, 256, kernel_size=(3, 3), stride=(1, 1), padding=(1, 1), groups=4, bias=False)
      (bn): BatchNorm2d(256, eps=1e-05, momentum=0.1, affine=True, track_running_stats=False)
      (lhc2): ConvAdapt(
        (gwc): Conv2d(256, 256, kernel_size=(3, 3), stride=(1, 1), padding=(1, 1), groups=32)
        (pwc): Conv2d(256, 256, kernel_size=(1, 1), stride=(1, 1))
      )
      (conv_expand): Conv2d(256, 512, kernel_size=(1, 1), stride=(1, 1), bias=False)
      (bn_expand): BatchNorm2d(512, eps=1e-05, momentum=0.1, affine=True, track_running_stats=False)
      (lhc3): ConvAdapt(
        (gwc): Conv2d(512, 512, kernel_size=(3, 3), stride=(1, 1), padding=(1, 1), groups=64)
        (pwc): Conv2d(512, 512, kernel_size=(1, 1), stride=(1, 1))
      )
    )
    (2): ResNeXtBottleneck(
      (conv_reduce): Conv2d(512, 256, kernel_size=(1, 1), stride=(1, 1), bias=False)
      (bn_reduce): BatchNorm2d(256, eps=1e-05, momentum=0.1, affine=True, track_running_stats=False)
      (conv_conv): Conv2d(256, 256, kernel_size=(3, 3), stride=(1, 1), padding=(1, 1), groups=4, bias=False)
      (bn): BatchNorm2d(256, eps=1e-05, momentum=0.1, affine=True, track_running_stats=False)
      (conv_expand): Conv2d(256, 512, kernel_size=(1, 1), stride=(1, 1), bias=False)
      (bn_expand): BatchNorm2d(512, eps=1e-05, momentum=0.1, affine=True, track_running_stats=False)
    )
  )
  (stage_3): Sequential(
    (0): ResNeXtBottleneck(
```

```
      (conv_reduce): Conv2d(512, 512, kernel_size=(1, 1), stride=(1, 1), bias=False)
      (bn_reduce): BatchNorm2d(512, eps=1e-05, momentum=0.1, affine=True, track_running_stats=False)
      (conv_conv): Conv2d(512, 512, kernel_size=(3, 3), stride=(2, 2), padding=(1, 1), groups=4, bias=False)
      (bn): BatchNorm2d(512, eps=1e-05, momentum=0.1, affine=True, track_running_stats=False)
      (conv_expand): Conv2d(512, 1024, kernel_size=(1, 1), stride=(1, 1), bias=False)
      (bn_expand): BatchNorm2d(1024, eps=1e-05, momentum=0.1, affine=True, track_running_stats=False)
      (downsample): Sequential(
        (0): Conv2d(512, 1024, kernel_size=(1, 1), stride=(2, 2), bias=False)
        (1): BatchNorm2d(1024, eps=1e-05, momentum=0.1, affine=True, track_running_stats=False)
      )
    )
    (1): ResNeXtBottleneck(
      (conv_reduce): Conv2d(1024, 512, kernel_size=(1, 1), stride=(1, 1), bias=False)
      (bn_reduce): BatchNorm2d(512, eps=1e-05, momentum=0.1, affine=True, track_running_stats=False)
      (conv_conv): Conv2d(512, 512, kernel_size=(3, 3), stride=(1, 1), padding=(1, 1), groups=4, bias=False)
      (bn): BatchNorm2d(512, eps=1e-05, momentum=0.1, affine=True, track_running_stats=False)
      (conv_expand): Conv2d(512, 1024, kernel_size=(1, 1), stride=(1, 1), bias=False)
      (bn_expand): BatchNorm2d(1024, eps=1e-05, momentum=0.1, affine=True, track_running_stats=False)
    )
    (2): ResNeXtBottleneck(
      (conv_reduce): Conv2d(1024, 512, kernel_size=(1, 1), stride=(1, 1), bias=False)
      (bn_reduce): BatchNorm2d(512, eps=1e-05, momentum=0.1, affine=True, track_running_stats=False)
      (conv_conv): Conv2d(512, 512, kernel_size=(3, 3), stride=(1, 1), padding=(1, 1), groups=4, bias=False)
      (bn): BatchNorm2d(512, eps=1e-05, momentum=0.1, affine=True, track_running_stats=False)
      (conv_expand): Conv2d(512, 1024, kernel_size=(1, 1), stride=(1, 1), bias=False)
      (bn_expand): BatchNorm2d(1024, eps=1e-05, momentum=0.1, affine=True, track_running_stats=False)
    )
  )
  (avgpool): AdaptiveAvgPool2d(output_size=(1, 1))
  (classifier): Linear(in_features=1024, out_features=100, bias=True)
)
```

ResNet-50 with adapters for ImageNet-to-ImageNetC

```
ResNetAdapt(
    (conv1): Conv2d(3, 64, kernel_size=(7, 7), stride=(2, 2), padding=(3, 3), bias=False)
    (bn1): BatchNorm2d(64, eps=1e-05, momentum=0.1, affine=True, track_running_stats=False)
    (relu): ReLU(inplace=True)
    (maxpool): MaxPool2d(kernel_size=3, stride=2, padding=1, dilation=1, ceil_mode=False)
    (layer1): Sequential(
      (0): BottleneckAdpt(
        (conv1): Conv2d(64, 64, kernel_size=(1, 1), stride=(1, 1), bias=False)
        (lhc1): ConvAdapt(
          (gwc): Conv2d(64, 64, kernel_size=(3, 3), stride=(1, 1), padding=(1, 1), groups=8)
          (pwc): Conv2d(64, 64, kernel_size=(1, 1), stride=(1, 1))
        )
        (bn1): BatchNorm2d(64, eps=1e-05, momentum=0.1, affine=True, track_running_stats=False)
        (conv2): Conv2d(64, 64, kernel_size=(3, 3), stride=(1, 1), padding=(1, 1), bias=False)
        (lhc2): ConvAdapt(
          (gwc): Conv2d(64, 64, kernel_size=(3, 3), stride=(1, 1), padding=(1, 1), groups=8)
          (pwc): Conv2d(64, 64, kernel_size=(1, 1), stride=(1, 1))
        )
        (bn2): BatchNorm2d(64, eps=1e-05, momentum=0.1, affine=True, track_running_stats=False)
        (conv3): Conv2d(64, 256, kernel_size=(1, 1), stride=(1, 1), bias=False)
        (bn3): BatchNorm2d(256, eps=1e-05, momentum=0.1, affine=True, track_running_stats=False)
        (relu): ReLU(inplace=True)
        (downsample): Sequential(
          (0): Conv2d(64, 256, kernel_size=(1, 1), stride=(1, 1), bias=False)
          (1): BatchNorm2d(256, eps=1e-05, momentum=0.1, affine=True, track_running_stats=False)
        )
      )
      (1): BottleneckAdpt(
        (conv1): Conv2d(256, 64, kernel_size=(1, 1), stride=(1, 1), bias=False)
        (lhc1): ConvAdapt(
          (gwc): Conv2d(64, 64, kernel_size=(3, 3), stride=(1, 1), padding=(1, 1), groups=8)
          (pwc): Conv2d(64, 64, kernel_size=(1, 1), stride=(1, 1))
        )
        (bn1): BatchNorm2d(64, eps=1e-05, momentum=0.1, affine=True, track_running_stats=False)
        (conv2): Conv2d(64, 64, kernel_size=(3, 3), stride=(1, 1), padding=(1, 1), bias=False)
        (lhc2): ConvAdapt(
          (gwc): Conv2d(64, 64, kernel_size=(3, 3), stride=(1, 1), padding=(1, 1), groups=8)
          (pwc): Conv2d(64, 64, kernel_size=(1, 1), stride=(1, 1))
        )
        (bn2): BatchNorm2d(64, eps=1e-05, momentum=0.1, affine=True, track_running_stats=False)
        (conv3): Conv2d(64, 256, kernel_size=(1, 1), stride=(1, 1), bias=False)
        (bn3): BatchNorm2d(256, eps=1e-05, momentum=0.1, affine=True, track_running_stats=False)
        (relu): ReLU(inplace=True)
      )
      (2): BottleneckAdpt(
        (conv1): Conv2d(256, 64, kernel_size=(1, 1), stride=(1, 1), bias=False)
        (lhc1): ConvAdapt(
          (gwc): Conv2d(64, 64, kernel_size=(3, 3), stride=(1, 1), padding=(1, 1), groups=8)
          (pwc): Conv2d(64, 64, kernel_size=(1, 1), stride=(1, 1))
        )
        (bn1): BatchNorm2d(64, eps=1e-05, momentum=0.1, affine=True, track_running_stats=False)
        (conv2): Conv2d(64, 64, kernel_size=(3, 3), stride=(1, 1), padding=(1, 1), bias=False)
        (lhc2): ConvAdapt(
          (gwc): Conv2d(64, 64, kernel_size=(3, 3), stride=(1, 1), padding=(1, 1), groups=8)
          (pwc): Conv2d(64, 64, kernel_size=(1, 1), stride=(1, 1))
        )
        (bn2): BatchNorm2d(64, eps=1e-05, momentum=0.1, affine=True, track_running_stats=False)
```

```
          (conv3): Conv2d(64, 256, kernel_size=(1, 1), stride=(1, 1), bias=False)
          (bn3): BatchNorm2d(256, eps=1e-05, momentum=0.1, affine=True, track_running_stats=False)
          (relu): ReLU(inplace=True)
        )
    )
    (layer2): Sequential(
      (0): BottleneckAdpt(
        (conv1): Conv2d(256, 128, kernel_size=(1, 1), stride=(1, 1), bias=False)
        (lhc1): ConvAdapt(
          (gwc): Conv2d(128, 128, kernel_size=(3, 3), stride=(1, 1), padding=(1, 1), groups=16)
          (pwc): Conv2d(128, 128, kernel_size=(1, 1), stride=(1, 1))
        )
        (bn1): BatchNorm2d(128, eps=1e-05, momentum=0.1, affine=True, track_running_stats=False)
        (conv2): Conv2d(128, 128, kernel_size=(3, 3), stride=(2, 2), padding=(1, 1), bias=False)
        (lhc2): ConvAdapt(
          (gwc): Conv2d(128, 128, kernel_size=(3, 3), stride=(1, 1), padding=(1, 1), groups=16)
          (pwc): Conv2d(128, 128, kernel_size=(1, 1), stride=(1, 1))
        )
        (bn2): BatchNorm2d(128, eps=1e-05, momentum=0.1, affine=True, track_running_stats=False)
        (conv3): Conv2d(128, 512, kernel_size=(1, 1), stride=(1, 1), bias=False)
        (bn3): BatchNorm2d(512, eps=1e-05, momentum=0.1, affine=True, track_running_stats=False)
        (relu): ReLU(inplace=True)
        (downsample): Sequential(
          (0): Conv2d(256, 512, kernel_size=(1, 1), stride=(2, 2), bias=False)
          (1): BatchNorm2d(512, eps=1e-05, momentum=0.1, affine=True, track_running_stats=False)
        )
      )
      (1): BottleneckAdpt(
        (conv1): Conv2d(512, 128, kernel_size=(1, 1), stride=(1, 1), bias=False)
        (lhc1): ConvAdapt(
          (gwc): Conv2d(128, 128, kernel_size=(3, 3), stride=(1, 1), padding=(1, 1), groups=16)
          (pwc): Conv2d(128, 128, kernel_size=(1, 1), stride=(1, 1))
        )
        (bn1): BatchNorm2d(128, eps=1e-05, momentum=0.1, affine=True, track_running_stats=False)
        (conv2): Conv2d(128, 128, kernel_size=(3, 3), stride=(1, 1), padding=(1, 1), bias=False)
        (lhc2): ConvAdapt(
          (gwc): Conv2d(128, 128, kernel_size=(3, 3), stride=(1, 1), padding=(1, 1), groups=16)
          (pwc): Conv2d(128, 128, kernel_size=(1, 1), stride=(1, 1))
        )
        (bn2): BatchNorm2d(128, eps=1e-05, momentum=0.1, affine=True, track_running_stats=False)
        (conv3): Conv2d(128, 512, kernel_size=(1, 1), stride=(1, 1), bias=False)
        (bn3): BatchNorm2d(512, eps=1e-05, momentum=0.1, affine=True, track_running_stats=False)
        (relu): ReLU(inplace=True)
      )
      (2): BottleneckAdpt(
        (conv1): Conv2d(512, 128, kernel_size=(1, 1), stride=(1, 1), bias=False)
        (lhc1): ConvAdapt(
          (gwc): Conv2d(128, 128, kernel_size=(3, 3), stride=(1, 1), padding=(1, 1), groups=16)
          (pwc): Conv2d(128, 128, kernel_size=(1, 1), stride=(1, 1))
        )
        (bn1): BatchNorm2d(128, eps=1e-05, momentum=0.1, affine=True, track_running_stats=False)
        (conv2): Conv2d(128, 128, kernel_size=(3, 3), stride=(1, 1), padding=(1, 1), bias=False)
        (lhc2): ConvAdapt(
          (gwc): Conv2d(128, 128, kernel_size=(3, 3), stride=(1, 1), padding=(1, 1), groups=16)
          (pwc): Conv2d(128, 128, kernel_size=(1, 1), stride=(1, 1))
        )
        (bn2): BatchNorm2d(128, eps=1e-05, momentum=0.1, affine=True, track_running_stats=False)
        (conv3): Conv2d(128, 512, kernel_size=(1, 1), stride=(1, 1), bias=False)
        (bn3): BatchNorm2d(512, eps=1e-05, momentum=0.1, affine=True, track_running_stats=False)
        (relu): ReLU(inplace=True)
      )
      (3): BottleneckAdpt(
        (conv1): Conv2d(512, 128, kernel_size=(1, 1), stride=(1, 1), bias=False)
        (lhc1): ConvAdapt(
          (gwc): Conv2d(128, 128, kernel_size=(3, 3), stride=(1, 1), padding=(1, 1), groups=16)
          (pwc): Conv2d(128, 128, kernel_size=(1, 1), stride=(1, 1))
        )
        (bn1): BatchNorm2d(128, eps=1e-05, momentum=0.1, affine=True, track_running_stats=False)
        (conv2): Conv2d(128, 128, kernel_size=(3, 3), stride=(1, 1), padding=(1, 1), bias=False)
        (lhc2): ConvAdapt(
          (gwc): Conv2d(128, 128, kernel_size=(3, 3), stride=(1, 1), padding=(1, 1), groups=16)
          (pwc): Conv2d(128, 128, kernel_size=(1, 1), stride=(1, 1))
        )
        (bn2): BatchNorm2d(128, eps=1e-05, momentum=0.1, affine=True, track_running_stats=False)
        (conv3): Conv2d(128, 512, kernel_size=(1, 1), stride=(1, 1), bias=False)
        (bn3): BatchNorm2d(512, eps=1e-05, momentum=0.1, affine=True, track_running_stats=False)
        (relu): ReLU(inplace=True)
      )
    )
    (layer3): Sequential(
      (0): BottleneckAdpt(
        (conv1): Conv2d(512, 256, kernel_size=(1, 1), stride=(1, 1), bias=False)
        (lhc1): ConvAdapt(
          (gwc): Conv2d(256, 256, kernel_size=(3, 3), stride=(1, 1), padding=(1, 1), groups=32)
          (pwc): Conv2d(256, 256, kernel_size=(1, 1), stride=(1, 1))
        )
        (bn1): BatchNorm2d(256, eps=1e-05, momentum=0.1, affine=True, track_running_stats=False)
        (conv2): Conv2d(256, 256, kernel_size=(3, 3), stride=(2, 2), padding=(1, 1), bias=False)
        (lhc2): ConvAdapt(
          (gwc): Conv2d(256, 256, kernel_size=(3, 3), stride=(1, 1), padding=(1, 1), groups=32)
          (pwc): Conv2d(256, 256, kernel_size=(1, 1), stride=(1, 1))
        )
```

```
      (bn2): BatchNorm2d(256, eps=1e-05, momentum=0.1, affine=True, track_running_stats=False)
      (conv3): Conv2d(256, 1024, kernel_size=(1, 1), stride=(1, 1), bias=False)
      (bn3): BatchNorm2d(1024, eps=1e-05, momentum=0.1, affine=True, track_running_stats=False)
      (relu): ReLU(inplace=True)
      (downsample): Sequential(
        (0): Conv2d(512, 1024, kernel_size=(1, 1), stride=(2, 2), bias=False)
        (1): BatchNorm2d(1024, eps=1e-05, momentum=0.1, affine=True, track_running_stats=False)
      )
    )
    (1): BottleneckAdpt(
      (conv1): Conv2d(1024, 256, kernel_size=(1, 1), stride=(1, 1), bias=False)
      (lhc1): ConvAdapt(
        (gwc): Conv2d(256, 256, kernel_size=(3, 3), stride=(1, 1), padding=(1, 1), groups=32)
        (pwc): Conv2d(256, 256, kernel_size=(1, 1), stride=(1, 1))
      )
      (bn1): BatchNorm2d(256, eps=1e-05, momentum=0.1, affine=True, track_running_stats=False)
      (conv2): Conv2d(256, 256, kernel_size=(3, 3), stride=(1, 1), padding=(1, 1), bias=False)
      (lhc2): ConvAdapt(
        (gwc): Conv2d(256, 256, kernel_size=(3, 3), stride=(1, 1), padding=(1, 1), groups=32)
        (pwc): Conv2d(256, 256, kernel_size=(1, 1), stride=(1, 1))
      )
      (bn2): BatchNorm2d(256, eps=1e-05, momentum=0.1, affine=True, track_running_stats=False)
      (conv3): Conv2d(256, 1024, kernel_size=(1, 1), stride=(1, 1), bias=False)
      (bn3): BatchNorm2d(1024, eps=1e-05, momentum=0.1, affine=True, track_running_stats=False)
      (relu): ReLU(inplace=True)
    )
    (2): BottleneckAdpt(
      (conv1): Conv2d(1024, 256, kernel_size=(1, 1), stride=(1, 1), bias=False)
      (lhc1): ConvAdapt(
        (gwc): Conv2d(256, 256, kernel_size=(3, 3), stride=(1, 1), padding=(1, 1), groups=32)
        (pwc): Conv2d(256, 256, kernel_size=(1, 1), stride=(1, 1))
      )
      (bn1): BatchNorm2d(256, eps=1e-05, momentum=0.1, affine=True, track_running_stats=False)
      (conv2): Conv2d(256, 256, kernel_size=(3, 3), stride=(1, 1), padding=(1, 1), bias=False)
      (lhc2): ConvAdapt(
        (gwc): Conv2d(256, 256, kernel_size=(3, 3), stride=(1, 1), padding=(1, 1), groups=32)
        (pwc): Conv2d(256, 256, kernel_size=(1, 1), stride=(1, 1))
      )
      (bn2): BatchNorm2d(256, eps=1e-05, momentum=0.1, affine=True, track_running_stats=False)
      (conv3): Conv2d(256, 1024, kernel_size=(1, 1), stride=(1, 1), bias=False)
      (bn3): BatchNorm2d(1024, eps=1e-05, momentum=0.1, affine=True, track_running_stats=False)
      (relu): ReLU(inplace=True)
    )
    (3): BottleneckAdpt(
      (conv1): Conv2d(1024, 256, kernel_size=(1, 1), stride=(1, 1), bias=False)
      (lhc1): ConvAdapt(
        (gwc): Conv2d(256, 256, kernel_size=(3, 3), stride=(1, 1), padding=(1, 1), groups=32)
        (pwc): Conv2d(256, 256, kernel_size=(1, 1), stride=(1, 1))
      )
      (bn1): BatchNorm2d(256, eps=1e-05, momentum=0.1, affine=True, track_running_stats=False)
      (conv2): Conv2d(256, 256, kernel_size=(3, 3), stride=(1, 1), padding=(1, 1), bias=False)
      (lhc2): ConvAdapt(
        (gwc): Conv2d(256, 256, kernel_size=(3, 3), stride=(1, 1), padding=(1, 1), groups=32)
        (pwc): Conv2d(256, 256, kernel_size=(1, 1), stride=(1, 1))
      )
      (bn2): BatchNorm2d(256, eps=1e-05, momentum=0.1, affine=True, track_running_stats=False)
      (conv3): Conv2d(256, 1024, kernel_size=(1, 1), stride=(1, 1), bias=False)
      (bn3): BatchNorm2d(1024, eps=1e-05, momentum=0.1, affine=True, track_running_stats=False)
      (relu): ReLU(inplace=True)
    )
    (4): BottleneckAdpt(
      (conv1): Conv2d(1024, 256, kernel_size=(1, 1), stride=(1, 1), bias=False)
      (lhc1): ConvAdapt(
        (gwc): Conv2d(256, 256, kernel_size=(3, 3), stride=(1, 1), padding=(1, 1), groups=32)
        (pwc): Conv2d(256, 256, kernel_size=(1, 1), stride=(1, 1))
      )
      (bn1): BatchNorm2d(256, eps=1e-05, momentum=0.1, affine=True, track_running_stats=False)
      (conv2): Conv2d(256, 256, kernel_size=(3, 3), stride=(1, 1), padding=(1, 1), bias=False)
      (lhc2): ConvAdapt(
        (gwc): Conv2d(256, 256, kernel_size=(3, 3), stride=(1, 1), padding=(1, 1), groups=32)
        (pwc): Conv2d(256, 256, kernel_size=(1, 1), stride=(1, 1))
      )
      (bn2): BatchNorm2d(256, eps=1e-05, momentum=0.1, affine=True, track_running_stats=False)
      (conv3): Conv2d(256, 1024, kernel_size=(1, 1), stride=(1, 1), bias=False)
      (bn3): BatchNorm2d(1024, eps=1e-05, momentum=0.1, affine=True, track_running_stats=False)
      (relu): ReLU(inplace=True)
    )
    (5): BottleneckAdpt(
      (conv1): Conv2d(1024, 256, kernel_size=(1, 1), stride=(1, 1), bias=False)
      (lhc1): ConvAdapt(
        (gwc): Conv2d(256, 256, kernel_size=(3, 3), stride=(1, 1), padding=(1, 1), groups=32)
        (pwc): Conv2d(256, 256, kernel_size=(1, 1), stride=(1, 1))
      )
      (bn1): BatchNorm2d(256, eps=1e-05, momentum=0.1, affine=True, track_running_stats=False)
      (conv2): Conv2d(256, 256, kernel_size=(3, 3), stride=(1, 1), padding=(1, 1), bias=False)
      (lhc2): ConvAdapt(
        (gwc): Conv2d(256, 256, kernel_size=(3, 3), stride=(1, 1), padding=(1, 1), groups=32)
        (pwc): Conv2d(256, 256, kernel_size=(1, 1), stride=(1, 1))
      )
      (bn2): BatchNorm2d(256, eps=1e-05, momentum=0.1, affine=True, track_running_stats=False)
      (conv3): Conv2d(256, 1024, kernel_size=(1, 1), stride=(1, 1), bias=False)
      (bn3): BatchNorm2d(1024, eps=1e-05, momentum=0.1, affine=True, track_running_stats=False)
```

```
            (relu): ReLU(inplace=True)
        )
    )
    (layer4): Sequential(
        (0): BottleneckAdpt(
            (conv1): Conv2d(1024, 512, kernel_size=(1, 1), stride=(1, 1), bias=False)
            (lhc1): ConvAdapt(
                (gwc): Conv2d(512, 512, kernel_size=(3, 3), stride=(1, 1), padding=(1, 1), groups=64)
                (pwc): Conv2d(512, 512, kernel_size=(1, 1), stride=(1, 1))
            )
            (bn1): BatchNorm2d(512, eps=1e-05, momentum=0.1, affine=True, track_running_stats=False)
            (conv2): Conv2d(512, 512, kernel_size=(3, 3), stride=(2, 2), padding=(1, 1), bias=False)
            (lhc2): ConvAdapt(
                (gwc): Conv2d(512, 512, kernel_size=(3, 3), stride=(1, 1), padding=(1, 1), groups=64)
                (pwc): Conv2d(512, 512, kernel_size=(1, 1), stride=(1, 1))
            )
            (bn2): BatchNorm2d(512, eps=1e-05, momentum=0.1, affine=True, track_running_stats=False)
            (conv3): Conv2d(512, 2048, kernel_size=(1, 1), stride=(1, 1), bias=False)
            (bn3): BatchNorm2d(2048, eps=1e-05, momentum=0.1, affine=True, track_running_stats=False)
            (relu): ReLU(inplace=True)
            (downsample): Sequential(
                (0): Conv2d(1024, 2048, kernel_size=(1, 1), stride=(2, 2), bias=False)
                (1): BatchNorm2d(2048, eps=1e-05, momentum=0.1, affine=True, track_running_stats=False)
            )
        )
        (1): BottleneckAdpt(
            (conv1): Conv2d(2048, 512, kernel_size=(1, 1), stride=(1, 1), bias=False)
            (lhc1): ConvAdapt(
                (gwc): Conv2d(512, 512, kernel_size=(3, 3), stride=(1, 1), padding=(1, 1), groups=64)
                (pwc): Conv2d(512, 512, kernel_size=(1, 1), stride=(1, 1))
            )
            (bn1): BatchNorm2d(512, eps=1e-05, momentum=0.1, affine=True, track_running_stats=False)
            (conv2): Conv2d(512, 512, kernel_size=(3, 3), stride=(1, 1), padding=(1, 1), bias=False)
            (lhc2): ConvAdapt(
                (gwc): Conv2d(512, 512, kernel_size=(3, 3), stride=(1, 1), padding=(1, 1), groups=64)
                (pwc): Conv2d(512, 512, kernel_size=(1, 1), stride=(1, 1))
            )
            (bn2): BatchNorm2d(512, eps=1e-05, momentum=0.1, affine=True, track_running_stats=False)
            (conv3): Conv2d(512, 2048, kernel_size=(1, 1), stride=(1, 1), bias=False)
            (bn3): BatchNorm2d(2048, eps=1e-05, momentum=0.1, affine=True, track_running_stats=False)
            (relu): ReLU(inplace=True)
        )
        (2): BottleneckAdpt(
            (conv1): Conv2d(2048, 512, kernel_size=(1, 1), stride=(1, 1), bias=False)
            (lhc1): ConvAdapt(
                (gwc): Conv2d(512, 512, kernel_size=(3, 3), stride=(1, 1), padding=(1, 1), groups=64)
                (pwc): Conv2d(512, 512, kernel_size=(1, 1), stride=(1, 1))
            )
            (bn1): BatchNorm2d(512, eps=1e-05, momentum=0.1, affine=True, track_running_stats=False)
            (conv2): Conv2d(512, 512, kernel_size=(3, 3), stride=(1, 1), padding=(1, 1), bias=False)
            (lhc2): ConvAdapt(
                (gwc): Conv2d(512, 512, kernel_size=(3, 3), stride=(1, 1), padding=(1, 1), groups=64)
                (pwc): Conv2d(512, 512, kernel_size=(1, 1), stride=(1, 1))
            )
            (bn2): BatchNorm2d(512, eps=1e-05, momentum=0.1, affine=True, track_running_stats=False)
            (conv3): Conv2d(512, 2048, kernel_size=(1, 1), stride=(1, 1), bias=False)
            (bn3): BatchNorm2d(2048, eps=1e-05, momentum=0.1, affine=True, track_running_stats=False)
            (relu): ReLU(inplace=True)
        )
    )
    (avgpool): AdaptiveAvgPool2d(output_size=(1, 1))
    (fc): Linear(in_features=2048, out_features=1000, bias=True)
  )
)
```

