# OpenReview forum: "FEATHER: Lifelong Test-Time Adaptation with Lightweight Adapters"
_ICLR.cc/2024/Conference — Submitted to ICLR 2024_

### Official Review · Reviewer_F1KC · 2023-10-16

**Soundness:** 3 good
**Presentation:** 2 fair
**Contribution:** 1 poor
**Rating:** 5
**Confidence:** 4

**Summary:**

This work tackles the problem lifelong Test-Time Adaptation (TTA): adapting on a sequence of domain shifts presented at test time. The authors propose FEATHER; an orthogonal approach to TTA methods in the literature.
Instead of adapting the weights of the pretrained model at test time, FEATHER inserts learnable adapters as additional modules in the network while keeping the original model parameters frozen.
The inserted adapters are initialized with identity mapping, so that the non-adapted model preserves its performance on source data.
Experiments are conducted on four standard TTA benchmarks: CIFAR10-C, CIFAR100-C, ImageNet-C, and ImageNet-3DCC.
The experimental results show that FEATHER achieves competitive performance compared to state-of-the-art methods while adapting smaller number of parameters.

**Strengths:**

This work has the following strengths:

- The problem this work tackles is both important and practical. Pretrained models are likely to experience domain shifts at test-time and adapting them on the fly is essential to ensure their reliability.

- The proposed approach is easy to understand and simple to implement.

- The experiments conducted in this work cover several standard benchmarks to prove the robustness of FEATHER.

**Weaknesses:**

Despite the strengths of this work, there are few weaknesses that should be addressed.

(1) FEATHER lacks the strong objective. The proposed approach seem not improve performance nor efficiency. I will explain next.
- (1a) From the performance perspective, FEATHER does not provide performance improvement over other baselines (e.g. CoTTA).
- (1b) From the efficiency perspective, while FEATHER updates a smaller percentage of network parameters when compared to CoTTA, FEATHER is less efficient than CoTTA. In essence, FEATHER adds extra parameters to the network making the forward pass more expensive. Further, the gradient calculation for conducting an update step of FEATHER and CoTTA are similar since FEATHER adds its adapters after every layer. Hence, the gradient will require a back propagation through the entire network.

(2) Missing experiments. While the experiments in this paper covered 4 benchmarks, there are key experiments missing to validate the effectiveness of FEATHER.
- (2a) Performance comparison. Strong and efficient baselines such as EATA, ECoTTA, and SAR shall be included in the main evaluation comparison. Appendix D only provides the comparison under one setup (ImageNet-C).
- (2b) Since FEATHER is an orthogonal approach to TTA methods, why is it assumed that it does not need accessing source data? For example, if FEATHER is combined with EATA, source data is necessary for the anti-forgetting regularizer. Having said that, I think it is necessary to compare FEATHER to TTA methods that leverage source data.
- (2c) Efficiency measures.  The efficiency comparison in this work is based on the percentage of parameters being updated compared to the total number of parameters. I am not sure if this is the right way to compare different TTA methods. First, FEATHER adds extra parameters to the network, and thus by construction, its forward pass is slower than the baseline (e.g. CoTTA). A comparison in terms of runtime and memory usage is more necessary. It is worth mentioning that methods like Tent and EATA only update the normalization layers making them even more efficient than FEATHER.

(3) Writing. The writing of this paper can be vastly improved in several places such as:
- The mathematical notation and problem description is not clear. In section 2 $f_\theta$ outputs the prediction $\hat y$ at the beginning, and later in the same paragraph it is assumed to output a probability vector.
- The notation in the last paragraph in Sections 3.1 and 3.2 are not clear. What is the element wise addition? how is that different from a regular addition? What is the input and output shapes of the adapting layers?
- Section 3.3 should mention that FEATHER leverages the same loss function as CoTTA. It was only clear in the experiments how exactly the adaptation is conducted.

**Questions:**

Suggestions: Here are some additional suggestions regarding the paper writing and organizing. Note that these comments were not taken into consideration in the paper evaluation.

- Both Figure 3 and section 3.2 are conveying a very simple message: Initialize the adapters with identity mapping. I would rather invest this space in more experiments with more insights (e.g. combining FEATHER with EATA).

- Formatting and writing: The proposed method is simple while the methodology section is not. I would try to simplify the writing of Section 3 and remove redundant paragraphs.

- Please consider reorganizing the tables in page 7 such that each table is presented with its own paragraph.

- For ImageNet-C experiments, please consider a similar setup to the CIFAR10/100-C experiments where the corruptions are ordered, similar to the continual evaluation in EATA.

---

> ### Author Response · Authors · 2023-11-20
> **Response to Reviewer F1KC**
>
> Thanks for your detailed review and positive remarks about our work tackling an important and practical problem, approach to be easy and simple to implement, and experimental evaluation on various benchmarks to be robust.
>
> We believe that there is some potential misunderstanding regarding some key aspects. We request the reviewer to take a look at our common response (and the revised version of the paper which incorporates the various suggestions). In addition, we also respond below to the reviewer’s individual comments. We hope our response will help address the questions/concerns and that the reviewer will reconsider the original assessment and consider raising the score.
>
> **Performance comparison of FEATHER vs. CoTTA:** Table 3 clearly demonstrates that, for ImageNet-to-ImageNetC, FEATHER, with only 10.92% trainable adapter parameters, outperforms CoTTA (which adapts all (100%) of the network parameters) in terms of error rate. Further, Table 4 shows that FEATHER’s performance is **comparable** to CoTTA and other approaches, even **with only 6.71%** trainable adapters, while **outperforming** all these methods **with as few as 50.32%** trainable parameters (Table 5). FEATHER can be used with any desired number of trainable parameters depending on the computational budget. In Table 7 in Appendix of the old version (Now Table 4 in the main paper), FEATHER with CoTTA-based TTA objective outperforms EcoTTA for both CIFAR10-to-CIFAR10C and ImageNet-to-ImageNetC datasets. For ImageNet-to-ImageNetC datasets, FEATHER outperforms all the other approaches, including EcoTTA and CoTTA.
>
> **Efficiency comparison of FEATHER vs CoTTA:** We believe the reviewer has missed a very important point here. The gradient calculation for conducting an update step of FEATHER and CoTTA is not similar. In FEATHER, the adapters are only present for some layers depending on the budget for the parameters. Thus, FEATHER does not introduce adapters after every layer. We have added the following line in section 3.1 to clarify this point: “In practice, we only insert the adapters in a few locations, depending on the computational and memory budget.”. Further, the **backpropagation only requires the computation of gradients of the adapter parameters and not the entire network.** (also see Appendix F for the architecture-specific adapter locations in detail.) Thus, FEATHER drastically improves the parameter efficiency over CoTTA. This is also reflected in the memory saving and wall-clock timing, which we have reported in the rebuttal (as requested by Reviewer CBjP) and in the updated version of the paper.
>
> **Comparison with baselines such as EATA, ECoTTA:** In Appendix Table 7 in the previous version (now Table 4 in the main paper), we have reported the results for the experiments on CIFAR10-to-CIFAR10C and ImageNet-to-ImageNetC, and FEATHER performs better than EcoTTA, which uses EATA loss.
>
> **Access to source data:** The initialization mechanism that we propose for FEATHER adapters circumvents the need for warm-up training of the adapters as in EcoTTA. Our approach does not inherently need source data for warm-up like EcoTTA. Our contribution is on the architectural aspect; however, if one wishes to use a learning objective like EATA, which depends on the source data for computing Fisher information, along with FEATHER, the source data would be utilized. However, this is a requirement of EATA and not that of FEATHER. We report the results of EATA + FEATHER in the updated version of the paper. (Table 6 and Table 7)
>
> **Efficiency (memory/run-time) comparisons:** We have added the comparison in terms of runtime and memory usage in our common response. We provide detailed reasons for comparing the methods based on the number of trainable parameters.  We believe there is some misunderstanding regarding the results reported for TENT (the entire network). We would like to highlight that the TENT (100% trainable) is an additional setting run by us for a transparent comparison. The original TENT method using only BN params performs poorly in the continual setting due to error accumulation and forgetting over time.

---

> > ### Author Response · Authors · 2023-11-20
> > **Response to Writing and Suggestions**
> >
> > **Writing:** Thanks for pointing out the places where you found the writing to be unclear. In particular, we would like to clarify that:
> > * By output, we meant the softmax probability vector (the predicted label corresponds to the index of the maximum value in the vector). We have clarified this.
> > * It should be an element-wise addition throughout; the regular addition is a typo. The output and input shapes of the adapter layers are equal and will depend on the location in the architecture where the adapter layers are added; for example, if the ith layer has an output shape of 32x32x15 and we add the adapter layer between ith and i+1th layer the adapter’s input and output shape would be 32x32x15, hence resulting in identity mapping of feature space during the initialization.
> > * We do not explicitly mention CoTTA loss in Section 3.3 because our approach is generic to be used with various losses. In the experiments, we utilize CoTTA loss.
> >
> >
> > **Other suggestions:**
> >
> > * Thanks for the suggestion about Figure 3 and section 3.2. We have updated the paper accordingly and have also added more experiments with FEATHER+EATA for comparison with EATA. Results are updated in Table 6.
> > * We have incorporated the writing and formatting suggestions in the updated version and have also adjusted the locations of the tables.
> > * In the updated version, we have included the results for ordered corruptions for ImageNet-C experiments (similar to EATA).
> > * The lifelong setting reported by EATA includes switching between the corruption and the source domain in an alternating fashion (please refer to https://github.com/mr-eggplant/EATA/blob/main/main.py#L123 for the official implementation), making it different from the followed lifelong definition of CTTA methods where the unseen domains (corruptions) are encountered in a continual fashion.  Nevertheless, for transparency, we report the FEATHER results with the EATA number taken from the paper directly (Table 11 in EATA paper). The comparison is presented below. Note that FEATHER shows an improvement over the average error rate, though the per corruption error rate reported by the EATA paper performs better, highlighting the overall effectiveness of FEATHER over EATA.
> >
> >
> > | Gauss. | Shot | Impul. | Defoc. | Glass | Motion | Zoom | Avg. |
> > |--------|------|--------|--------|-------|--------|------|------|
> > | $34.9_{ \pm 0.2}$ | $36.9_{ \pm 0.1}$ | $35.8_{ \pm 0.2}$ | $33.6_{ \pm 0.3}$ | $33.3_{ \pm 0.2}$ | $47.2_{ \pm 0.3}$ | $52.7_{ \pm 0.1}$ | $35.8_{ \pm 0.2}$ |
> > | $26.58_{ \pm 0.07}$ | $25.25_{ \pm 0.079}$ | $28.94_{ \pm 0.068}$ | $24.35_{ \pm 0.08}$ | $22.57_{ \pm 0.06}$ | $35.91_{ \pm 0.07}$ | $41.61_{ \pm 0.03}$ | $37.36_{ \pm 0.02}$ |

---

> > ### Author Response · Authors · 2023-11-22
> > **Last day for the Discussion Period**
> >
> > Dear Reviewer F1KC,
> >
> > As the author-reviewer discussion period is about to get over (Nov 22, end-of-day AoE time), we request you to please take a look at our author response (and the updated version of the paper, which incorporates your suggestions and includes the experiments you had suggested along with the additional comparisons) and let us know it your concerns are addressed, and if you have any follow-up questions.
> >
> > Thanks.

---

### Official Review · Reviewer_M9Xj · 2023-10-30

**Soundness:** 2 fair
**Presentation:** 2 fair
**Contribution:** 2 fair
**Rating:** 5
**Confidence:** 4

**Summary:**

This work presents a novel FEATHER method for lifelong/continual test-time adaptation problems. With the lightweight adapter and freezing the base model, FEATHER is able to adapt the source pre-trained model to the non-stationary test distributions without forgetting the source knowledge and eliminate the error accumulation. More specifically, FEATHER inserts learnable adapter in to the source pre-trained model and only updates them with the unlabeled test data. And the work designs zero and identity initialization for adapters to preserving source knowledge. Experiments show that the FEATHER can achieve comparable performance with SOTA by adjusting few parameters.

**Strengths:**

1. Leveraging adapters to address catastrophic forgetting and reduce the accumulation of errors is well motivated.
2. The paper is well organized.

**Weaknesses:**

1. How to determine where to insert adapter？In the work, a combination of PWC and GCO servers as a basic adapter. And adapters will be inserted in to the model between layers. However, there is no mention of where in the network the adapter should be inserted which is one of the most important aspects of method based on adapters. Where to add, from shallow layer to deep layer, or from deep layer to shallow layer, there is no specific experiment to analyze.
2. Why ZERO AND IDENTITY INITIALIZATION preserves the source knowledge? Maybe a residue architecture with zero initialization will be more simple and effective?
3. Difference with ECoTTA [1]. In fact, adding adapters between layers has already been proposed in ecotta, and in their analysis experiments, they tried a similar variation of the method in their paper (refer to Architecture design in Section 4.2 in ECoTTA [1]). The choice and novelty of adapter structure is still open to question.
4. What we should be noticed is that parameter efficient does not mean resource efficient! Inserting adapter into the original model will leads more computational resource including FLOPTS and memory of GPU. But there was little or no performance improvement.
5. Over claim on the combination of other methods. Combined with tent and cotta, there is no performance improvement.
6. What is the objective function in the experiments of Table 1,2,3,4,5?
7. The formulation of Lifelong/Continual Test-time adaptation have some inconformity with existing literature. In CoTTA [2], it is define as adapting the model to the test samples and make predictions for them in an online manner. The description in section 2 seems like an offline fashion.

Refecrence:
[1] Song et al., Ecotta: Memory-efficient continual test-time adaptation via self-distilled regularization. In CVPR, 2023.
[2] Wang et al., Continual test-time domain adaptation. In CVPR 2022.

**Questions:**

Please refer to [Weakness].

---

> ### Author Response · Authors · 2023-11-20
> **Response to Reviewer M9Xj**
>
> We thank the reviewer for the comments and suggestions. We request the reviewer to take a look at the common response and the updated version of the paper, which incorporates your and other reviewers’ suggestions. Below, we provide some clarifications regarding the results and novelty of the proposed approach. We hope that you will consider revising your assessment and raising your score.
>
> **Deciding where to add adapters:** Our approach based on adding adapters in between the layers is generic. Based on the computational budget of the deployment environment, the adapters can be added to the main architecture. In the updated version of the paper, we report the locations of the added adapter layers in Appendix F, and the configs for all the architectures will be released with the code base. The location of adapter parameters is dependent on the TTA environment. For example, if the expected noise is corruption, the domain shifts are expected in the initial layers, whereas if the domain shifts occur in semantic space, the suitable adapter locations would be in the deep(er) layers (close to the output layer). In the TTA setting, all the information about the domain shifts is usually not available a priori. To be generic and comply with the TTA setting, this decision should be made based on the dataset and the architecture for which FEATHER is being used. In our experiments and lifelong TTA settings, it would be unfair to predefine/suggest/assume any details related to the domain shifts. Hence the location of the used adapters is generic. (As described in Appendix F)
>
> **Zero and Identity initialization vs residue architecture:** The zero and Identity initialization preserves source knowledge since it acts as an identity operation to begin with during the start. Even though a residue architecture with zero initialization is simpler, it is not as effective as our proposed PWC and GCO-based adapters. In our earlier experiments, we found that, when adding a residual layer, the network collapses when the domain changes occur in the continual setting and adapter parameters are updated. We speculate the primary reason to be the flow of signal to pass through the added parameters and, after the update, the network starts to collapse. We also tried some other initialization schemes like near zero init; however, those do not work well with the continual TTA setup.
>
> **Difference with EcoTTA:** As compared to EcoTTA, FEATHER employs more sophisticated adapters, which is supported by better performance in continual TTA, shown in Table 4 (Table 7 in the previous version), where we compare with EcoTTA as well. Please refer to general comments for more discussion and comparison with EcoTTA. Also, one may refer to section 4.2 of the EcoTTA paper for architecture-specific details, which are quite different from FEATHER.
>
> **Savings in terms of computational/memory requirements:** Thanks for pointing this out and making this suggestion. We report the memory and compute savings in the general response and have updated the paper to also include these results. Our approach is more resource-efficient in terms of computational resources such as memory and FLOPs, which are usually required during the backward passes done during adaptation in test time. Further, we perform better than EcoTTA and other SOTA approaches in continual TTA, as highlighted in Table 4.
>
> **Improvements when combining FEATHER with other methods:** We respectfully disagree that there is no improvement when FEATHER is combined with methods such as TENT and CoTTA. As compared to standard TENT and CoTTA, which require updating all the parameters (100%) during adaptation,  when these methods are augmented with FEATHER, significantly fewer parameters need to be updated, as shown in Table 6. Note that (as we mention in our common response as well as reported by the additional experiments) a reduction in trainable parameters directly affects the memory budget requirement as well as inference/adaptation time  (Table 7).
>
> **The objective function:** Thank you for pointing this out. The objective function in the experiments of Tables 1,2,3,4,5 is the same as that of CoTTA. We briefly mention this in the second line of Section 5.2. In the updated version of the paper, we highlight this in the table captions as well.
>
> **Inconformity in the formulation of Lifelong/Continual TTA:** Note that, in Section 2, we start by defining the standard (non-continual) TTA setting. However, in the second last paragraph, we define the continual/lifelong TTA setting. Since the primary focus of this work is the behavior of adapters in a continual setting, we follow the continual setup for the entire experiments and results section and report the observed findings.

---

> ### Author Response · Authors · 2023-11-22
> **Last day for the Discussion Period**
>
> Dear Reviewer M9Xj,
>
> As the author-reviewer discussion period is about to get over (Nov 22, end-of-day AoE time), we request you to please take a look at our author response (and the updated version of the paper, which incorporates your suggestions and includes the experiments you had suggested) and let us know it your concerns are addressed, and if you have any follow-up questions.
>
> Thanks.

---

### Official Review · Reviewer_bPzr · 2023-10-30

**Soundness:** 2 fair
**Presentation:** 2 fair
**Contribution:** 2 fair
**Rating:** 3
**Confidence:** 4

**Summary:**

This paper proposes an adapter-based method for lifelong test-time adaptation. The authors assume that the given source model is a CNN and insert adapters composed of group convolutions and 1x1 convolutions between the layers of the model. During test time, only the inserted adapter and the batch normalization (BN) parameters of the source model are updated, while the remaining weights are fixed. The experiments demonstrate that the proposed approach achieves performance comparable to state-of-the-art methods, despite updating a very small number of trainable parameters.

**Strengths:**

1. This paper is overall clearly clarified and well organized.
2. The use of adapters for lifelong test-time adaptation seems novel.

**Weaknesses:**

1. The proposed method is only applicable to CNNs.
2. The authors argue for the importance of preserving the information of the source model in online TTA tasks. However, the proposed method does not exhibit outstanding performance compared to the CoTTA, which involves full fine-tuning of the model. Therefore, the authors fail to sufficiently explain why the use of adapters is suitable for online TTA tasks, apart from the fact that it reduces the number of trainable parameters.
3. The authors claim that the proposed method excels in terms of parameter update costs; however, the cost of training the adapters inserted between the model's layers, in terms of memory and computation, is not significantly lighter compared to full fine-tuning.

**Questions:**

1. Is it possible to apply the proposed method to Transformers?
2. Why does CoTTA generally outperform the proposed method?
3. Could you compare the proposed method with other methods in terms of memory and computation?

---

> ### Author Response · Authors · 2023-11-20
> **Response to Reviewer bPzr**
>
> We thank the reviewer for the comments and suggestions, and are glad that the reviewer finds our idea novel. Please refer to our common response as well as the updated version of the paper that has incorporated the various comments from all the reviewers. Below, we respond to your specific comments/concerns, and we hope that you will consider revising your opinion and consider increasing the score in light of our response.
>
> **Is FEATHER only applicable to CNNs?** As we mention in the common response, although we demonstrate our idea for generic CNN-based architectures for computer vision, our idea is general and can be applied to any deep learning architecture (including transformers) that allows the insertion of adapters between layers. Some implementational aspects will need to be changed based on the underlying architecture, but the rest of the idea is more broadly applicable.
>
> **Improvement over CoTTA:** Please refer to our common response.
>
> **Memory and computation savings:** Please refer to our common response.
>
> **Why CoTTA “outperforms” our method?** We respectfully disagree with this and would like to draw your attention to a few key points (also mentioned in our common response). Please note that, while CoTTA with **all** parameters trainable outperforms FEATHER marginally, CoTTA also has significantly more trainable parameters. In contrast, FEATHER only uses ~7.2% to 12.2% trainable parameters and yet performs comparably to CoTTA. Further, we demonstrate in Table 5 that with only 50.32% trainable parameters, FEATHER **outperforms** CoTTA, which highlights that even adapting just around half the number of parameters is enough.

---

> ### Author Response · Authors · 2023-11-22
> **Last day for the Discussion Period**
>
> Dear Reviewer bPzr,
>
> As the author-reviewer discussion period is about to get over (Nov 22, end-of-day AoE time), we request you to please take a look at our author response (and the updated version of the paper, which incorporates your suggestions and includes some additional experiments) and let us know it your concerns are addressed, and if you have any follow-up questions.
>
> Thanks.

---

> > ### Comment · Reviewer_bPzr · 2023-11-22
> > **Post-rebuttal**
> >
> > Thanks to the authors for addressing my concerns. However, I still have some concerns regarding the following points:
> > 1. Of course, one can imagine applying a similar approach to different architectures, but I believe it is meaningful to actually implement and observe the effects. As far as I know, TTA experiments do not incur high costs, yet despite that, the authors do not provide experimental results for different architectures.
> > 2. The authors emphasize the importance of preserving source knowledge in TTA scenarios, proposing a method that freezes the source model and relies on adapters, but it does not surpass the performance of a full fine-tuning approach. There is a need to deliver contributions beyond confirming observations from existing adapter studies in the TTA scenario.
> > 3. The reduction in memory or computation cost does not proportionally align with the reduction in parameters. To enhance TTA efficiency, there are better adapter-based methods available [1, 2].
> >
> > [1] Zhang, Jeffrey O., et al. "Side-tuning: a baseline for network adaptation via additive side networks.", ECCV 2020.
> > [2] Sung, Yi-Lin, Jaemin Cho, and Mohit Bansal. "Lst: Ladder side-tuning for parameter and memory efficient transfer learning.", NeurIPS 2022.

---

> ### Author Response · Authors · 2023-11-22
>
> Thank you for engaging in a discussion. We believe there is some potential misunderstanding about some of the key aspects, and we request you to look at our response below, which we hope will clarify the misunderstanding and we hope that you can do a fair assessment in light of these points.
>
> 1. For our experiments, for a fair comparison and to assess the improvement of performance over existing methods, we follow the architectures proposed in existing methods on lifelong TTA on the popularly used RobustBench platform, where the architectures are CNN-based. Since the numbers reported by other methods in the lifelong TTA setting use such architectures, it would be unfair to compare them against another architecture, like a transformer-based architecture.
>
> 2. We would reiterate that our aim is not to surpass the performance of the full fine-tuning approach but rather to perform at least comparably with a significantly smaller number of parameters (which also translates into memory and time savings, as our results show). Our experimental results show that FEATHER is flexible in terms of the available computational budget and even outperforms CoTTA if the computational budget is increased (roughly 50% of CoTTA budget; please refer to Table 5). Please note the primary aim here is to reduce the computational budget. Moreover, the results in Table 3 for the ImageNet-to-ImageNetC dataset show that FEATHER achieves an improvement with only ~10% of parameters being trained.
>
> 3. We respectfully disagree that a reduction in memory or computation cost does not align with the reduction in parameters (as shown in Table 7 and explained in the text around it, we get improvements on these metrics, too).  We would also like to highlight that **both the papers mentioned by you are about standard fine-tuning for transfer learning and not for TTA or lifelong TTA**, whereas our work is about lifelong TTA. Lifelong TTA presents a very different set of challenges than standard fine-tuning, which is done during the training phase, whereas TTA or lifelong TTA must perform fine-tuning at inference time using the unlabeled test input(s).
>
> We hope our response above clarifies the potential misunderstandings and that you reconsider your original assessment of our paper. We would be happy to answer any other concerns/comments you might have.

---

> > ### Comment · Reviewer_bPzr · 2023-11-23
> >
> > Thank you for the comment. My last question is:
> > 1. How does the proposed adapter differ from previously suggested adapters in transfer learning research, and why is it superior in TTA scenarios?

---

> ### Author Response · Authors · 2023-11-23
>
> Thanks for going through our earlier response and the follow-up question. Our response to your question is given below, and we hope that it helps clarify the **difference** between adapter design and **update strategies used in lifelong TTA** and **standard transfer learning**.
>
> The adapters used in our work are designed specifically, keeping in mind the challenges posed by TTA and lifelong TTA. Standard adapters used in transfer learning are not directly applicable in TTA (and especially lifelong TTA) because:
>
> (1) In TTA and lifelong TTA, the adapter parameters are trained at inference time using an **unsupervised loss** (which can be CoTTA, TENT, EATA, etc.), whereas, in transfer learning, during fine-tuning on the target domain, the adapter parameters are **trained using a supervised loss** defined on the target domain's labeled examples. Because of the need for unsupervised adapter parameter updates, we leverage learning objectives that are based on entropy or cross-entropy (utilizing pseudo labels).
>
> (2) Unlike standard transfer learning, in lifelong TTA, due to the need for unsupervised finetuning, there is a potential risk of error accumulation over longer horizons as the test domains keep drifting over time. Our framework FEATHER, because of the design strategies, such as **identity initialization** of the adapter, ensures that such error accumulation is mitigated effectively.
>
> (3) In transfer learning using adapters, the adapter updates can leverage the labeled examples for multiple iterations, whereas, in our lifelong TTA setting, we have to operate in an **online** setting where we do not have the luxury to revisit the unlabeled test samples more than once. In this aspect, too, FEATHER’s adapter update strategy is different from the adapter update used by standard supervised fine-tuning-based transfer learning.

---

### Official Review · Reviewer_CBjP · 2023-11-01

**Soundness:** 2 fair
**Presentation:** 3 good
**Contribution:** 2 fair
**Rating:** 5
**Confidence:** 4

**Summary:**

The paper focuses on the continual test time adaptation (CTTA), where the test input has a time-varying domain. The authors propose cost-effective CTTA method, called FEATHER, which mitigates the forgetting issue of previous methods in CTTA scenarios. The proposed method is particularly useful in practice as it does not require any access to the source dataset, whereas most of the existing ones do. The authors demonstrate the efficacy of the proposed method in various CTTA scenarios. In addition, they empirically show a comparative advantage over existing methods, even including ones requiring access to the source domain in a TTA scenario.

**Strengths:**

The proposed method, FEATHER, addresses the forgetting issue in CTTA with no access to the source domain, which is often limited in the practice of TTA yet is required for existing methods. Specifically, it proposes to employ a set of new parameters and clever initialization, that do not harm the performance at the beginning (without seeing the source dataset), while enabling an effective prevention of the forgetting issue. There was a similar approach (EcoTTA: Song et al. 2023) but it requires warm-up phase to find such a harmless initialization of additional parameters based on the source dataset. In addition, FEATHER is memory-efficient as it reduces the number of parameters to be updated in the procedure of TTA. Such benefit and efficacy of the proposed method have been demonstrated on a set of experiments (CIFAR10C, CIFAR100C, ImageNetC, ImageNet3DCC).

**Weaknesses:**

My major concern is the limited justification of the proposed method. In my understanding, the main selling point is to address CTTA problems at reduced computational complexity with no access to the training dataset. However, there seem no comparisons to SOTA algorithms in terms of CTTA performance and computational complexity, although they require some access to the training dataset. In addition, as the reduced number of updating parameters does not always imply reduced computational complexity, it is necessary to report time complexity (wall-clock time to process a batch). Lastly, the proposed method may have architecture-dependent effectiveness. Hence, it is also necessary to provide discussion or experiment with various model architectures.

**Questions:**

In my understanding, Table 7 reports the performance on the most basic TTA scenario (i.e., no continual setup). Please clarify the setting for Table 7. If my understanding is correct, please provide a fair comparison to existing state-of-the-art (SOTA) methods in CTTA, although SOTA methods need some access to the training dataset. This would help to understand the effectiveness of the proposed method.

The proposed method seems an architecture-specific solution. Is it possible to apply FEATHER to other model architectures (e.g., ViT or other CNN-based ones)?

Can you provide a comparison of TTA methods in terms of time complexity (wall-clock time) per batch? I do understand the time complexity would be proportional to the number of parameters. However, there can be other computational cost in TTA algorithms. For instance, in my understanding, COTTA is particularly slow due to the use of data augmentation to obtain a more robust pseudo-label.

In my understanding, it is straightforward to make a variant of COTTA, which updates only BN layers. Noting that adapting BN layers is parameter-effective in TTA, it is also interesting to compare FETHER to the COTTA variant, in terms of parameter complexity and TTA/CTTA performance.

---

> ### Author Response · Authors · 2023-11-20
> **Response to Reviewer CBjP**
>
> Thanks for your detailed and insightful comments. In addition to our common response to all the reviewers, we provide below our response to your specific concerns/questions and hope that you will reconsider your original assessment and revise your score as you deem appropriate.
>
> **Comparison with other SOTA methods that use source data:** Please see our common response.
>
> **Memory usage and wall-clock time comparisons:** In our common response, we have explained why we had shown results about savings in the number of parameters and how it results in reductions in memory usage and wall-clock time. We agree with your suggestion that it will be insightful to report these numbers as well, and have performed additional experiments and included these results in the updated version of the paper (Table 7). The table clearly shows that reducing the parameters (specifically _**updatable parameters**_) directly reduces the memory consumption by a significant margin, as well as the time required for the adaptation. Note that the reported wall-clock times are averaged over all 15 corruptions for all the methods.
>
> **Is the approach architecture-specific?** Please see our common response.
>
> **Clarification about Table 7:** We apologize for the misunderstanding. The experiments in Table 7 (of the original paper), as well as the rest of the paper, follow a **lifelong/continual TTA** setup (not standard TTA). We have updated the caption in Table 7 in the new version of the paper (Table 4 in the updated version of the paper).
>
> **Variant of CoTTA with only BN params updates:** Thanks for your suggestion. In the updated version, we report results (Table 10) not just for CoTTA with BN params but also for EATA (BN params), EATA (all params), and EATA + FEATHER (adapter params). Our experiments highlight that, with a significantly smaller number of trainable parameters, similar performance can be achieved while keeping the source knowledge intact.

---

> ### Author Response · Authors · 2023-11-22
> **Last day for the Discussion Period**
>
> Dear Reviewer CBjP,
>
> As the author-reviewer discussion period is about to get over (Nov 22, end-of-day AoE time), we request you to please take a look at our author response (and the updated version of the paper, which incorporates your suggestions and includes the experiments you had suggested) and let us know it your concerns are addressed, and if you have any follow-up questions.
>
> Thanks.

---

### Author Response · Authors · 2023-11-20
**Common Response to the Reviewers**

We thank the reviewers for their insightful comments and valuable feedback. The reviewers have found our problem setup to be well-motivated with practical scenarios and important for real-world development (Reviewer F1KC), novel in terms of usage of adapters for continual TTA setting (Reviewer bPzr), and effective in reducing the computational complexity with no access to the training dataset (Reviewer CBjP). As remarked by (Reviewer F1KC), the proposed idea of adding additional adaptable parameters with no updates to pre-trained weights is novel and easy to implement.

In this common response, we first discuss some of the main points raised by the reviewers; in particular (1) whether the parameter-efficiency also translates to memory-efficiency and wall-clock time savings, (2) comparison with other recent SOTA methods, and (3) applicability of our method to other architectures. In addition to our response, we also request the reviewers to take a look at the updated version of the paper, which includes the additional experiments that we conducted based on the reviewers' suggestions (for easy comparison/reference, the text color is blue for the updated parts).

**Parameter-efficiency vs memory consumption and wall-clock timing:** When the number of trainable parameters is reduced, it directly affects the _**memory usage**_ (required for maintaining parameter-specific gradients and activation maps for computing those gradients) as well as the _**adaptation time**_ (as gradients need to be computed with respect to each learnable parameter at inference time). To confirm if parameter-efficiency translates to memory savings and wall-clock time savings (at inference time) in practice, we performed additional experiments and have included their results in the updated version of the paper. In Table 7 and Appendix Table 10 of the updated paper, we show these results for FEATHER applied on three TTA objectives -- CoTTA, TENT, and EATA, and compare it with two test-time adaptation schemes: (1) adapting by updating only BN parameters ("BN Params"), and (2) adapting by updating all the params ("All Params"). Our results in Table 7 clearly show that FEATHER (regardless of the wrapper TTA method it is used with, be it CoTTA, TENT, or EATA) provides an advantage of adapting a minuscule percentage of params to achieve error rates that are comparable to full-finetuning ("All Params"), and much better error rates as compared to methods that adapting only batch-norm parameters ("BN Params") (Table 6 and Appendix Table 10), making it more practical and flexible for real-life deployment of the TTA models.

**Comparison with other SOTA methods (both that use source data and those that are source-free):** In Appendix D of our original submission, we had already provided comparisons with other recent SOTA methods like EATA and EcoTTA. In the updated version, these results can be found in Table 4 (with source data dependency). FEATHER outperforms both types of methods. Note that, on CIFAR10-To-CIFAR10C, CoTTA (which requires updating all the parameters) does marginally better than FEATHER, but FEATHER is much better than CoTTA in terms of parameter-efficiency as well as memory and wall-clock time (Table 7).

**Applicability of our method to other architectures:** Although the implementational details will require some changes (we focused on CNN-based architectures), our idea is general and can be used with any architecture that uses adapter modules between layers (including variants of transformers that use adapter modules).

We now highlight some of the key aspects that distinguish FEATHER from other recent approaches.

- A key advantage of FEATHER is that it does not require access to source domain data, unlike other recent SOTA methods like EATA and EcoTTA. Thus, it perfectly fits the problem of test-time or source-free domain adaptation, where the source data is no longer available when adapting to unlabeled test data. The source-free setting is more challenging and appealing to real-world applications where data privacy is the primary concern and crucial. Hence, TTA, which only requires access to the source model with a lifelong setup, directly applies to model deployment in the wild. In this work, we specifically aim to reduce the parameter update cost by proposing a new scheme, FEATHER, and show that a similar lifelong TTA performance can be achieved with a huge reduction in parameter update cost for the existing lifelong TTA methods.

- Another advantage of FEATHER is its ability to increase the expressive power of the network based on the computational budget. As our experiments show, even under a very tight computational budget (number of adapter params), FEATHER outperforms other recent parameter-efficient SOTA methods, and as the number of trainable params (added adapter params) is increased, FEATHER is competitive to CoTTA (which requires full fine-tuning) while still requiring a much smaller number of trainable params.

---

### Meta-Review · Area_Chair_tbBc · 2023-12-15

**Metareview:**

FEATHER is a method for test-time adaptation (TTA), and specifically continual test-time adaptation, in which the model must update online in order to generalize better to shifted data. FEATHER does not update the source model weights, but instead adds new "adapters" to the model and only updates their parameters, and takes care to initalize these adapters to the identity function for stable optimization during testing. Its proposed contributions are the design of the adapter, the choice of where to include the adapter(s) in the source model, the identity initialization of the adapters, and experiments to measure the accuracy and computation needed by FEATHER vs. existing TTA methods. It is noteworthy that the choice of adapter parameterization makes FEATHER compatible with a variety of TTA methods and losses: the experiments cover a number of entropy minimization methods (TENT, CoTTA, EATA) but do not include self-supervised methods (like contrastive learning, auxiliary tasks, ...). Results are shown on the standard TTA benchmarks of ImageNet-C and CIFAR-10/100-C as well as the newer 3D corruptions of ImageNet-3DCC: on accuracy FEATHER rivals or exceeds the  continual TTA state-of-the-art CoTTA and on computation it almost halves memory to 58% of CoTTA and reduces time to 90% of CoTTA.

The strengths are that FEATHER adapters are simple, that FEATHER is compatible with and complementary to a variety of TTA methods, and that FEATHER reduces computation while mostly maintaining accuracy w.r.t. the state-of-the-art for continual TTA. The weaknesses are that the adapters are established by prior work (Houlsby et al. 2019 as cited, thought the setting differs, and [AdaptFormer NeurIPS'22](https://proceedings.neurips.cc/paper_files/paper/2022/hash/69e2f49ab0837b71b0e0cb7c555990f8-Abstract-Conference.html), which is in the same setting but uncited), that the computational improvements are not more significant (at ~50-90% of the full-update computation of CoTTA) and more computation than TENT, and that the accuracy difference is marginal.

All reviewers vote for rejection with borderline (CBjP, M9Xj, F1KC) and reject (bPzr) ratings. The AC acknowledges the respectful and detailed author comment. The AC confirms that discrepancies between review content and submission content have been discounted, in particular the true/false facts of whether there is improvement to performance and efficiency, but must note that there is valid disagreement about the magnitude or significance of the differences. Having closely read the submission itself, the AC downweights the weaknesses raised by bPzr and the reject rating accordingly, because the limitation to CNNs and the priority of adapters from transfer learning do not hold. Nevertheless, there is consensus among reviewers for borderline rejection and the AC ultimately does not find sufficient grounds to overrule the consensus. The authors are encouraged to resubmit their work incorporating the improvements from this round of review. In that direction, reviewer F1KC acknowledged the improvement of the rebuttal during the reviewer-AC discussion and raised their score to 5, but indicated that more analysis and ablation of the use of adapters is needed to be thoroughly informative.

Note: Although concurrent work was not factored into the decision to ensure fairness (while ICLR'23 is in scope, it is close the cutoff in the 2024 policy), the AC would like to highlight related work on memory-efficient TTA in case it is of interest to the authors. See [MECTA: Memory-Economic Continual Test-Time Model Adaptation](https://openreview.net/forum?id=N92hjSf5NNh) at ICLR'23.

**Justification For Why Not Higher Score:**

While there is a degree of improvement in the experiments and information for a test-time adaptation audience, the significance of the results and the novelty w.r.t. prior work has not reached the bar for ICLR.

- Novelty is Limited by Prior Adapters: AdaptFormer at NeurIPS'22 already introduced test-time adapters and requires a reference. While there are differences, in that the adapters were applied to ViTs rather than CNNs and their initialization was near-identity rather than exactly the identity, these differences need to be addressed and justified by FEATHER.
- Efficiency in Parameters, Time, and Memory: FEATHER does not impart sufficient time or memory efficiency. To be clear, the use of adapters does reduce the number of parameters, time, and memory w.r.t. full updates. However, it is not competitive with BN updates in efficiency (Table 7) although it does improve in accuracy. All-in-all, FEATHER thus offers more flexibility in trading off computation and accuracy but does not strictly dominate existing methods.
- Reset Cost: The claimed improved practicality for resetting only holds for full-update methods like CoTTA and EcoTTA. Note that TENT need only keep the BN parameters and statistics for resetting, and not the full model parameters, so its memory cost is indeed less than FEATHER (as indicated in Table 7 of the submission: TENT updates 0.37% of the parameters while FEATHER updates 6.80% in its adapters).

**Justification For Why Not Lower Score:**

N/A

---

### Decision · Program_Chairs · 2024-01-16

Reject